# EXO1 as a therapeutic target for Fanconi Anaemia, ZRSR2 and BRCA1-A complex deficient cancers

Marija Maric [1], Sandra Segura-Bayona [1], Raviprasad Kuthethur [2], Tohru Takaki[1], Valerie Borel[1], Tyler H. Stanage [1], Miroslav P. Ivanov [1], Nishita Parnandi[1], Graeme Hewitt[1,5], Rhona Millar[1], Carmen S. Fonseca[2], Harshil Patel [3], Miriam Llorian [3], Scott Warchal[4], Michael Howell [4], Arnab Ray Chaudhuri[2], Panagiotis Kotsantis [1,6] & Simon J. Boulton [1] ✉

Exonuclease EXO1 performs multiple roles in DNA replication and DNA damage repair (DDR). However, EXO1 loss is well-tolerated, suggesting the existence of compensatory mechanisms that could be exploited in DDR-deficient cancers. Using CRISPR screening, we find EXO1 loss as synthetic lethal with many DDR genes somatically inactivated in cancers, including Fanconi Anaemia (FA) pathway and BRCA1-A complex genes. We also identify the spliceosome factor and tumour suppressor ZRSR2 as synthetic lethal with loss of EXO1 and show that ZRSR2-deficient cells are attenuated for FA pathway activation, exhibiting cisplatin sensitivity and radial chromosome formation. Furthermore, FA or ZRSR2 deficiencies depend on EXO1 nuclease activity and can be potentiated in combination with PARP inhibitors or ionizing radiation. Finally, we uncover dysregulated replication-coupled repair as the driver of synthetic lethality between EXO1 and FA pathway attributable to defective fork reversal, elevated replication fork speeds, post-replicative single stranded DNA exposure and DNA damage. These findings implicate EXO1 as a synthetic lethal vulnerability and promising drug target in a broad spectrum of DDR-deficient cancers unaddressed by current therapies.

DNA damage repair (DDR) pathways have emerged as important targets for cancer therapy. Germline mutations in DDR factors have been recognised amongst common cancer driver mutations, as exemplified by BRCA and mismatch repair gene mutations in homologous recombination (HR) deficient and microsatellite unstable tumours, respectively[1–3]. Cancer genomic studies have also revealed frequent somatic mutations in DDR pathways, which may enhance mutation rates and drive genome instability, which could fuel tumour evolution[4]. However, approved and developing therapeutic strategies[5] only target a subset of recognised DDR deficiencies in cancers, including PARP inhibitors and WRN loss in homologous recombination and mismatch repair deficient cancers, respectively, highlighting the need to identify novel therapeutic targets. Loss of a DDR pathway often confers a critical dependency on alternative repair pathways to maintain cell viability. Hence, defining synthetic lethal interactions provides a means to identify targets for such dependencies.

[1]DSB Repair Metabolism Laboratory, The Francis Crick Institute, London, UK. [2]Department of Molecular Genetics, Erasmus Cancer Institute, Erasmus University Medical Center, Rotterdam, The Netherlands. [3]Bioinformatics and Biostatistics Facility, The Francis Crick Institute, London, UK. [4]High Throughput Screening Facility, The Francis Crick Institute, London, UK. [5]Present address: Genome Stability Group, Comprehensive Cancer Centre, School of Cancer and Pharmaceutical Sciences, King's College London, London, UK. [6]Present address: Division of Biomedical and Life Sciences, Faculty of Health and Medicine, Lancaster University, Lancaster, UK. ✉e-mail: simon.boulton@crick.ac.uk

Here we define synthetic lethal interactions with exonuclease EXO1, which has revealed known and previously unappreciated genetic vulnerabilities that exist in many cancers. Specifically, we uncover cancer vulnerabilities amongst synthetic lethal hits with EXO1 loss, including deficiencies in BRCA1-A complex factors, FA pathway genes and the splicing factor and tumour suppressor ZRSR2. We show that ZRSR2 loss phenocopies FA pathway deficiency and implicate EXO1 and FANCG in replication fork reversal and the suppression of post-replicative ssDNA gaps.

## Results

### Identification of cancer vulnerabilities synthetic lethal with loss of EXO1

EXO1 is a conserved multi-functional exonuclease that performs roles in mismatch repair[6,7], homologous recombination[8–10], nucleotide excision repair[11,12], translesion DNA synthesis[13], resection of DSBs and stalled replication forks[14] and processing of Okazaki fragments[15]. Despite its multiple roles in DNA replication and repair, EXO1 loss is well tolerated as exemplified by the viability of Exo1 knockout mice and EXO1 KO cell lines[16–18]. However, Project Achilles[19], available through the DepMap portal, which collates information on cancer dependencies, defines EXO1 as a 'strongly selective' gene, indicating that EXO1 is essential in a subset of cancers. Based on these observations, we sought to test the hypothesis that EXO1 participates in compensatory pathways that DDR-deficient cancer cells critically rely on for viability. To this end, we generated constitutive knockout (KO) cell lines for EXO1 in diploid chronic myelogenous leukaemia eHAP and in cervical adenocarcinoma HeLa Kyoto cells harbouring an integrated inducible Cas9 (Fig. 1a, Supplementary Fig. 1a). Consistent with Exo1 KO mice and previously reported EXO1 KO cells[17,18], eHAP iCas9 and HeLa Kyoto iCas9 EXO1 KO cell lines generated using two different sgRNAs retain cell viability (Fig. 1b, Supplementary Fig. 1b–d). We functionally validated EXO1 KO cells by testing sensitivity to DNA damaging agents and inhibitors of DDR factors (Supplementary Fig. 1e). We found that eHAP iCas9 EXO1 KO clones are sensitive to PARP inhibitor olaparib, DNA crosslinking agents cisplatin and mitomycin C (MMC), ionising radiation, oxidating agent potassium bromate, alkylating agent methylmethanesulfonate (MMS), and the DNA–protein crosslinker formaldehyde. eHAP iCas9 EXO1 KO clones are also sensitive to the ribonucleotide reductase inhibitor hydroxyurea (HU) and inhibitor of the replication stress checkpoint kinase, ATR (Fig. 1c–e, Supplementary Fig. 1f–k). Olaparib and MMS sensitivities were also observed in EXO1 KO clones in HeLa Kyoto iCas9 cells (Supplementary Fig. 1l-m). Hence, loss of EXO1 is well tolerated in eHAP and HeLa Kyoto cancer cell lines and KOs exhibit previously observed sensitivities to different DNA damaging agents and inhibition of DDR factors[10,20,21,22,23], reflecting the multiple roles of EXO1 in DDR mechanisms.

Next, we performed a genome-wide CRISPR dropout screen using the Brunello sgRNA library in eHAP iCas9 wild-type and EXO1 KO cell lines and compared the levels of sgRNAs by deep sequencing in the two isogenic cell lines 6 and 16 days after induction of Cas9 expression (Fig. 1f, Supplementary Fig. 1n, o, Supplementary Data 1 and 2). sgRNAs significantly reduced in the EXO1 KO cells relative to wild type were categorised as synthetic lethal with loss of EXO1. Notably, GO Biological Process analysis of the screen data identified DDR pathways as the most significantly enriched processes among the top synthetic lethal hits (Fig. 1g). Further analysis revealed three distinct synthetic lethal categories with EXO1 loss: (1) genes previously shown to be synthetic lethal with EXO1 in model organisms or human cells, (2) DDR genes not previously linked to EXO1 loss that are frequently mutated in cancers, and (3) genes frequently mutated in cancer that have not been implicated in genome stability maintenance processes (Fig. 1h–j). The identification of multiple previously reported EXO1 synthetic lethal genetic interactions from model organism studies[24,25] confirmed the validity and success of our screen. Among these genes were FEN1

nuclease, which processes Okazaki fragments during DNA replication, MRE11A and RAD50, which form the essential MRN nuclease complex required for sensing and short-range resection of DNA double-strand breaks (DSB), BLM, TOP3A and RMI2, which form the BTRR complex required for long-range resection of DSBs and Holliday junction dissolution, and ATM kinase, which coordinates the cellular responses to DSBs (Fig. 1i, j).

We confirmed the synthetic lethal interaction between EXO1 and the FEN1 nuclease by demonstrating sensitivity of EXO1 KO cell lines to a FEN1 inhibitor (Supplementary Fig. 2a). Wild type and EXO1 KO cell lines carrying an integrated sgRNA for the BTRR complex factor RMI2 were tested for viability using either a clonogenic assay or a viability assay based on ATP levels. Induction of RMI2 KO in EXO1 KO cells following addition of doxycycline resulted in a significant loss of viability in the EXO1 KO cells relative to the isogenic wild-type control (Supplementary Fig. 2b–f, Supplementary Data 3). It should be noted that with an ~90% Cas9 cutting activity in the eHAP iCas9 cells, ~10% of the viable cells are unedited and should be considered when evaluating the synthetic lethal interaction in this and other assays. In agreement with published genome-wide CRISPR screens in lung cancer cell lines with an ATM kinase inhibitor[25], we confirmed the sensitivity of eHAP EXO1 KO cell lines to an ATM kinase inhibitor (Supplementary Fig. 2g). Together, these results demonstrate evolutionary conservation of synthetic lethal interaction with EXO1 loss from budding yeast to humans.

### EXO1 is synthetic lethal with FA pathway genes and BRCA1-A complex genes, which are somatically mutated in cancers

We next examined EXO1 synthetic lethal interactions with genome stability maintenance factors with frequent loss-of-function mutations or deletions in cancers. Amongst the most significant synthetic lethal hits are genes that constitute the FANCM complex (FANCM, FAAP24/C19orf40, APITD1/MHF1/CENPS, and STRA13/MHF2/CENPX), the Fanconi Anaemia (FA) core complex (FANCB, FANCC, FANCE, FANCF, FANCG, FANCL, FAAP20) and downstream effectors of the FA pathway, including FANCD2 (Fig. 1i, j). FA genes have been identified based on germline biallelic mutations that are present in Fanconi Anaemia patients, which are characterised, among other features, by bone marrow failure and predisposition to cancers[26]. Mechanistic studies have implicated the FA pathway in the DNA replication stress response and DNA crosslink repair. Initial DNA lesion recognition by the FANCM complex is followed by the recruitment of the FA core complex, which acts as an E3 ubiquitin ligase required to monoubiquitylate FANCD2-FANCI on chromatin to induce the response[27]. Heterozygous germline mutations of FA genes, which include FA core complex genes and FANCM, have been associated with predisposition to cancers, while homozygous deletions, somatic mutations and gene silencing of FA genes are also a common feature of many cancers[28]. Data from cancer genomic studies[29] show that somatic homozygous deletions in FA core or FANCM complex genes leading to loss of function are found at a frequency of 3.42% across all cancer types within the current TCGA cohort of 10,953 patients from 32 studies. Furthermore, a recent ICGC/TCGA pancancer analysis of whole genomes detected homozygous deletions of FANCB caused by a retrotransposition event on the X chromosome[30,31] in 6.74% of 2658 whole cancer genomes (Table 1). Somatic mutations in FA core complex and FANCM complex factors are also found in 6.58% of all cancer samples from the TCGA cohort, with a significant proportion harbouring two or more mutations, allowing for potential compound heterozygous effects (Table 1).

We also identified BRCA1-A complex factors (FAM175A/ABRAXAS1, BRCC3/BRCC36, and BABAM1/MERIT40) as synthetic lethal hits with loss of EXO1 (Fig. 1i, j). EXO1 was previously found to be essential for the viability of BRCA1-deficient cells[32], which was attributed to EXO1's role in DSB repair via the single-strand annealing pathway[33]. Notably, loss of BRCA1 in our CRISPR screen led to loss of viability both in wild

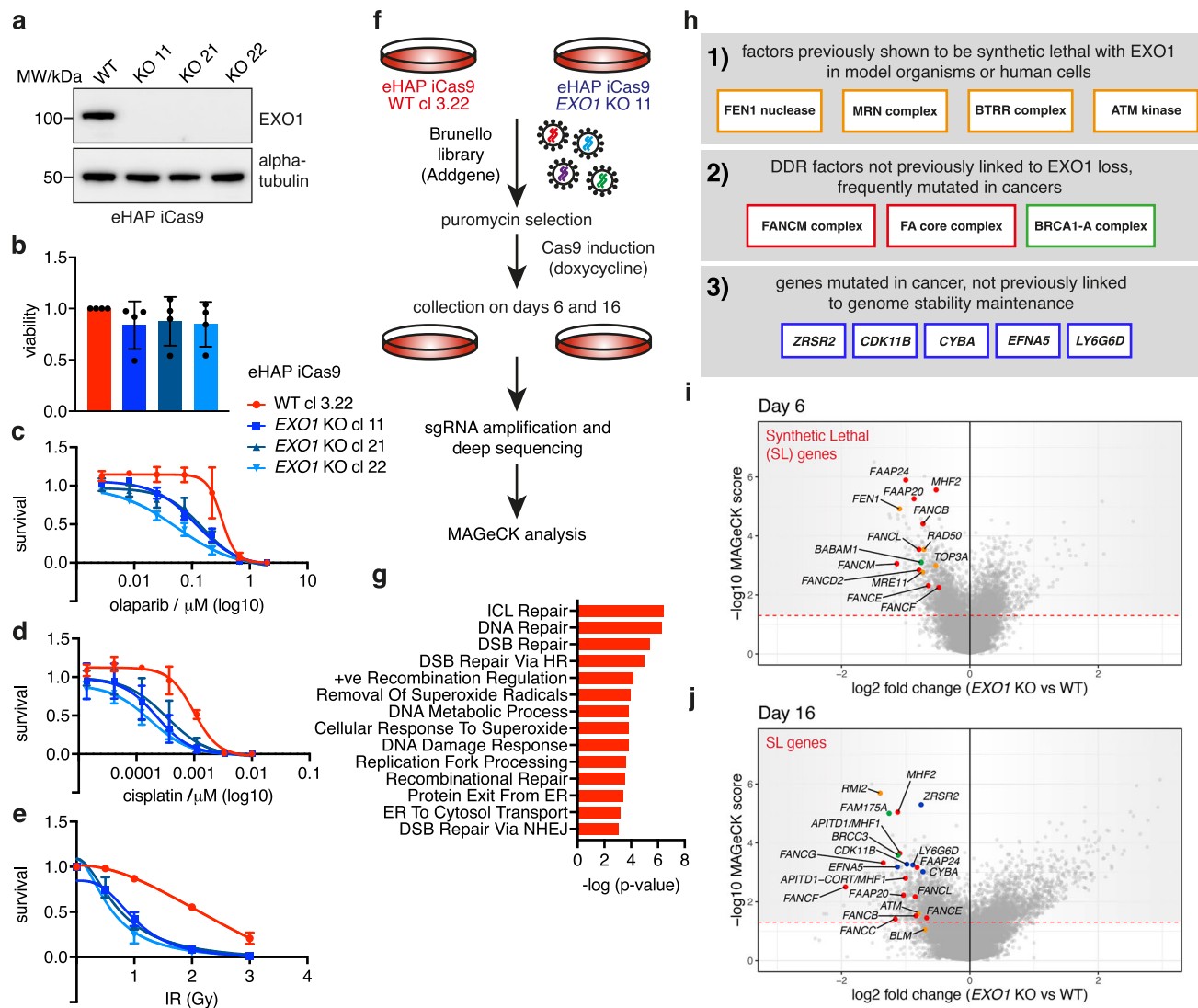

**Fig. 1 | Identification of cancer vulnerabilities synthetic lethal with loss of EXO1.**
**a** Immunoblotting for EXO1 protein levels in eHAP iCas9 *EXO1* KO clones (alpha-tubulin as control for equal loading; *n* = 2; MW molecular weight). **b** Viability of eHAP iCas9 *EXO1* KO clones based on CellTitre-Glo assay (*n* = 4), mean with standard deviation (SD). **c** Sensitivity of eHAP iCas9 *EXO1* KO clones to olaparib (*n* = 3, mean and error with SD). **d** Sensitivity of eHAP iCas9 *EXO1* KO clones to cisplatin (*n* = 3, mean and error with SD). **e** Sensitivity of eHAP iCas9 *EXO1* KO clones to ionising radiation (IR) (*n* = 3, mean and error with SD). **f** Scheme of genome-wide CRISPR dropout screen in eHAP iCas9 wild type and *EXO1* KO isogenic pair of cell lines. **g** GO Biological Process (2023) term analysis for top synthetic lethal hits from the screen. Based on log fold change and significance parameters, 547 genes were selected from both time points. Significance of the enrichment for top GO terms is represented in a graph for −log of *P* value, which was calculated with Fisher's exact test. **h** Overview of relevant categories of synthetic lethal hits. Three categories of genes of interest are described in separate boxes, with their identity listed in the respective box and denoted with the same colour as is used for the gene on the volcano plots in panels (**i**) and (**j**). **i** Volcano plot of 'day 6' comparison of *EXO1* KO vs. WT cells. Data was plotted based on parameters from the CRISPR screen MAGeCK analysis (log2 fold change on the *X*-axis and −log10 MAGeCK score on the *Y*-axis), with genes of interest highlighted as in (**h**). **j** Volcano plot of 'day 16' comparison of *EXO1* KO vs. WT cells. Equivalent to the comparison described in (**i**). Source data for **a**–**e** are provided as a Source Data file.

type and in *EXO1* KO cells due to its essentiality, while loss of its interacting partner *BARD1* was found as synthetic lethal with EXO1 loss (Supplementary Data 1). Given that BRCA1-A complex is one of three distinct BRCA1 complexes and its factors scored among the top synthetic lethal hits in *EXO1* KO CRISPR dropout screen, we focused on the characterisation of these synthetic lethal interactions. BRCA1-A complex is required for the recognition of DSBs through ubiquitylated histones, and when absent, results in an over-resection phenotype in HR-dependent DSB repair[34–38]. Like BRCA1, BRCA1-A complex factors are tumour suppressor genes[39], with homozygous deletions in the genes forming the complex estimated at a frequency of 1.04% across all cancer types within the current TCGA cohort of 10,953 patients, and with an additional 1.69% of all cancers harbouring mutations (Table 1). Furthermore, the *BRCC3* gene encoding for the catalytic subunit of

BRCA1-A complex is homozygously deleted in 5.40% of 2658 whole cancer genomes analysed by ICGC/TCGA due to an Xp22.2-linked retrotransposition event[30,31] (Table 1).

Based on their high significance in the screen as well as prevalence of loss-of-function somatic mutations in cancers, we performed validation of EXO1 synthetic lethal interactions with FANCM complex factors *FAAP24* and *APITD1*, FA core complex factor *FANCG*, and BRCA1-A complex factors *FAM175A* and *BRCC3* using clonogenic and viability assays. We observed significant reductions in viability for each of the induced double eHAP iCas9 KO cell lines (Fig. 2a–c, Supplementary Fig. 3, Supplementary Data 3). Inducible double KO HeLa Kyoto iCas9 cells also showed a significant reduction in cell viability with the clonogenic assay method (Supplementary Fig. 4a–i). Furthermore, we tested the impact of *EXO1* KO in the patient-derived

**Table 1 | Curated data from TCGA studies (10,967 samples from 10,953 patients from 32 studies) and ICGC/TCGA pancancer analysis of whole genomes (2683 samples from 2565 patients) for homozygous deletions and mutations**

| Gene | Hom. deletions in TCGA cohort (10,967 samples) (%) | Mutations in TCGA cohort (%) | Additive percentage for complexes/categories in TCGA cohort | Cumulative percentage for complexes/categories in TCGA cohort | Cumulative percentage of hom. deletions in TCGA cohort | Hom. deletions in ICGC/ TCGA pancancer analysis of whole genomes (2683 samples) (%) | Cumulative percentage of hom. deletions in ICGC/TCGA pancancer analysis of whole genomes (2683 samples) |
|---|---|---|---|---|---|---|---|
| FANCA | 1.17 | 1.65 | | | | 1.49 | |
| FANCB | 0.62 | 1.21 | | | | 6.74 | |
| FANCC | 0.25 | 0.84 | | | | 0.19 | |
| FANCE | 0.05 | 0.54 | | | | 0.03 | |
| FANCF | 0.08 | 0.39 | | | | 0.03 | |
| FANCG | 0.07 | 0.68 | Hom. deletions: 4.22%, Mutations: 10.45% | Hom. deletions 3.42%, Mutations 6.58% | | 0.26 | |
| FANCL | 0.07 | 0.62 | | | | 0.03 | |
| FAAP20 | 0.79 | 0.18 | | | | 0.48 | |
| FAAP100 | 0.14 | 0.91 | | | | 0.26 | |
| FANCM | 0.18 | 2.80 | | | | 0.11 | |
| FAAP24 | 0.13 | 0.41 | | | | 0.34 | |
| APITD1 | 0.47 | 0.16 | | | 4.38% | 0.37 | 8.49% |
| STRA13 | 0.20 | 0.06 | | | | 0.33 | |
| FAM175A | 0.21 | 0.44 | | | | 0.29 | |
| BRCC3 | 0.42 | 0.56 | Hom. deletions: 1.03%, Mutations 2.03% | Hom. deletions: 1.04%, Mutations: 2.03% | | 5.40 | |
| BABAM1 | 0.11 | 0.35 | | | | 0.22 | |
| RAP80 | 0.29 | 0.68 | | | | 0.22 | |
| ZRSR2 | 0.59 | 0.47 | | | | 6.63 | |
| CDK11B | 0.79 | 0.90 | | | | 0.48 | |
| CYBA | 1.09 | 0.23 | Hom. deletions: 3.48%, Mutations 2.14% | Hom. deletions: 3.24%, Mutations: 1.86% | | 1.34 | |
| EFNA5 | 0.87 | 0.35 | | | | 0.89 | |
| LY6G6D | 0.14 | 0.19 | | | | 0.03 | |

Data from cBioPortal as of March 2024. 'Additive' refers to the sum of individual values for each gene, while 'cumulative' refers to the percentage derived for the entire group when interrogating cBioPortal with the respective gene list.

FANCC-deficient pancreatic cancer cell line PL11[40], which has a defect in FANCD2 monoubiquitylation that can be restored by complementation with wild-type FANCC (Supplementary Fig. 4j). While *EXO1* KO induction did not have a significant effect on the PL11 cell line complemented with wild-type FANCC, the FANCC-deficient PL11 cell line showed significant growth reduction upon *EXO1* KO induction (Supplementary Fig. 4k–n).

To investigate how the induction of KOs of FA genes and BRCA1-A genes in EXO1-deficient cells compromises viability, we utilised a time-resolved cell proliferation assay coupled with an apoptosis-activated DNA-binding dye. Firstly, the cell proliferation assay confirmed that double KO cells (here shown through the example of *EXO1-FANCG* loss) show a significantly reduced proliferation profile in comparison to controls (Supplementary Fig. 5a, b). Coupling this assay with the apoptosis tracker, we observed an increase in apoptosis for *EXO1* and *FANCG* double KO cells (Fig. 2d), as well as for double KO cells of *EXO1* and FANCM complex factors (*FAAP24* and *APITD1*), and BRCA1-A complex factor *FAM175A* (Supplementary Fig. 5c–e), demonstrating that the observed synthetic lethal phenotypes trigger cell death by apoptosis, which is a critical finding in defining the suitability of EXO1 as a target in cancer therapy.

We next examined whether the observed cell death in double KO cells is preceded by increased genome instability. First, we performed cell cycle analysis using EdU analogue incorporation as a proxy for DNA synthesis and DAPI staining as a proxy for DNA content. We observed that inducible double KOs of *EXO1* and *FANCG* showed a significant increase in cells in the G2 phase of the cell cycle relative to wild-type controls. G2 phase arrest is a well-established phenotype of cells experiencing genome instability (Fig. 2e,

Supplementary Fig. 5f, g). We further validated the genome instability phenotype that occurs prior to cell death by analysing DDR markers through high content imaging and immunoblotting. We observed significant increases for the genome instability marker micronuclei over time in double KO cells of *EXO1* in combination with FA factors (*FANCG*, *FAAP24*, and *APITD1*) and BRCA1-A complex factors (*FAM175A* and *BRCC3*) (Fig. 2f, Supplementary Fig. 5h, i). We also observed significant increases of foci of phosphorylated RPA at serine 33, a marker of replication stress (Fig. 2g, Supplementary Fig. 5i). Immunoblotting further confirmed increases in DSB signalling markers, shown here for *EXO1-FANCG* double KO cells, through ATM-dependent phosphorylation of KAP1 (on serine 824), p53 (on serine 15) and CHK2 (on threonine 68), as well as enrichment of phosphorylated histone variant H2AX (on serine 139; gamma-H2AX) on chromatin (Fig. 2h). Altogether, these results suggest that inactivation of both EXO1 and FA pathway or BRCA1-A complex genes leads to accumulation of DNA damage and cell death.

## Loss of the spliceosome factor and tumour suppressor ZRSR2 phenocopies FA pathway deficiency and is synthetic lethal with loss of EXO1

Next, we analysed factors that have a significant negative genetic interaction with EXO1 loss but have not been previously implicated in maintaining genome stability. These factors were further triaged based on cancer genomic data for those frequently deleted in cancers (Table 1). Based on this analysis we focused on the spliceosome factor ZRSR2, cyclin-dependent kinase CDK11B, NAPDH oxidase factor CYBA, ephrin ligand EFNA5 and LY6G6D, a factor predicted as a member of the lymphocyte antigen complex (Fig. 1j). For these factors we

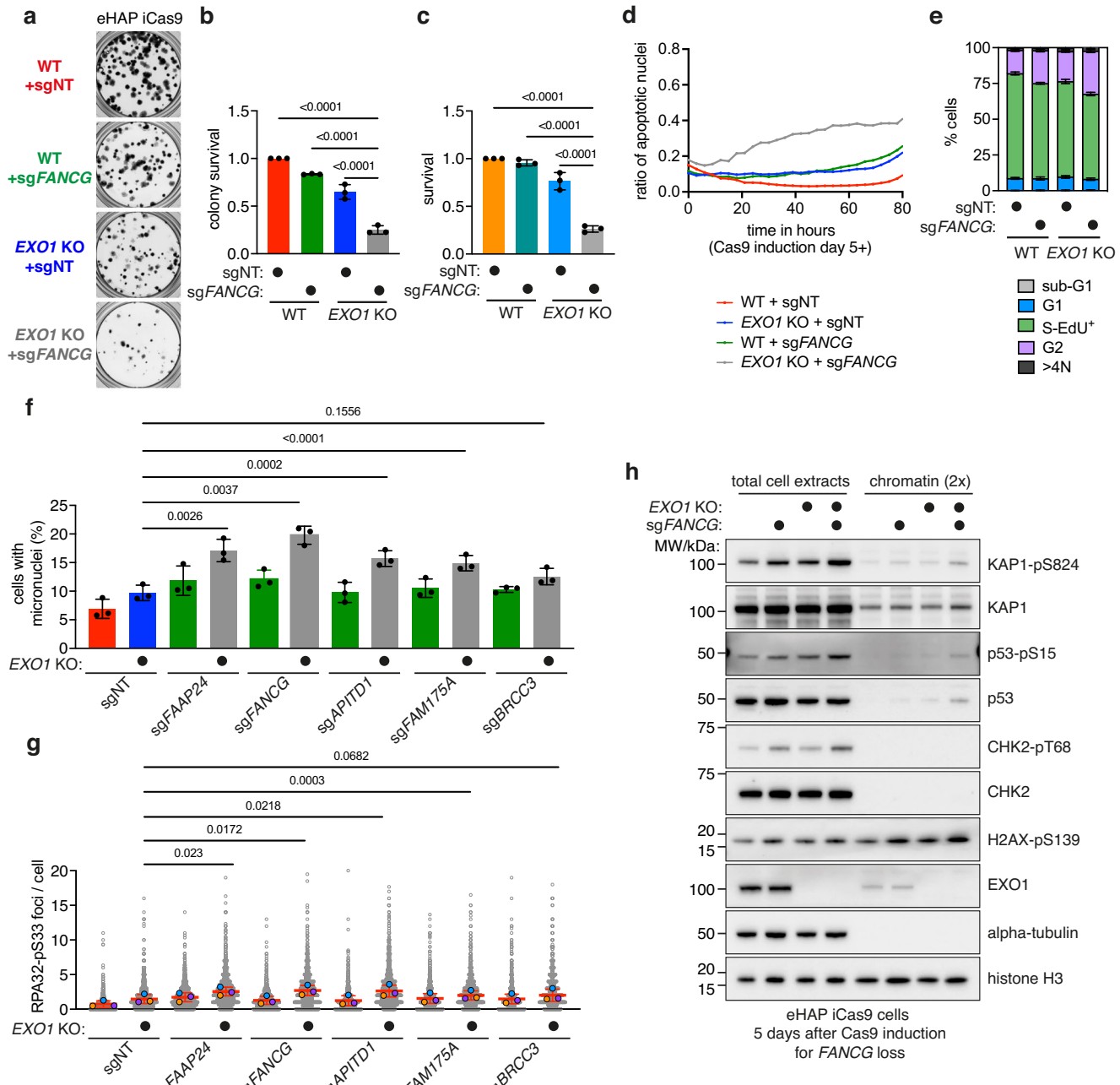

**Fig. 2 | EXO1 is synthetic lethal with FA pathway genes and BRCA1-A complex genes, which are somatically mutated in cancers. a** Validation of *EXO1* and *FANCG* synthetic lethal interaction in eHAP iCas9 cells by clonogenic assay (representative images, *n* = 3, well diameter 15.5 mm). **b** Quantification of clonogenic assay (*n* = 3, mean with SD) in (**a**) with ordinary one-way ANOVA statistical analysis (*F* = 158.8). **c** Validation of *EXO1* and *FANCG* synthetic lethal interaction in eHAP iCas9 cells by CellTitre-Glo viability assay (*n* = 3, mean with SD) with ordinary one-way ANOVA statistical analysis (*F* = 132.5). **d** Apoptosis of *EXO1* and *FANCG* double KO cells (*n* = 3). **e** Quantification of cell cycle phases (*n* = 4, based on gating shown in Supplementary Fig. 5g, mean with SD). **f** Micronuclei increase in double KO cells. Paired *t*-test analyses were performed for each of the double KO to compare to '*EXO1* KO + sgNT' (*n* = 3, mean with SD, with each percentage calculated on the basis of >2181 nuclei;

two-tailed *P* values). **g** Increase in phosphorylated RPA at serine 33 (RPA-pS33) foci in double KO cells. Dots in orange, blue and green represent the mean of a biological repeat, with the red line as their mean. Paired *t*-test analyses were performed on the means of biological repeat for each of the double KO cell lines in comparison to the '*EXO1* KO + sgNT' cell line (*n* = 3, mean with SD, with each percentage calculated on the basis of >754 nuclei; RPA-pS33 signal was measured within nuclei; two-tailed *P* values). **h** Immunoblotting of total cell extracts and chromatin for 'WT + sgNT', 'WT + sg*FANCG*', '*EXO1* KO + sgNT' and '*EXO1* KO + sg*FANCG*' eHAP iCas9 cells at day 5 post Cas9 expression induction. Samples were probed for phosphorylated levels of KAP1, p53, CHK2 and H2AX with controls, and EXO1, alpha-tubulin and histone H3 to control for sample identity and equal loading (representative experiment, *n* = 3). Source data for **b**–**h** are provided as a Source Data file.

confirmed the findings from the CRISPR screen using the clonogenic assay in the eHAP iCas9 cell line, which showed a significant reduction in viability in each of the induced double KO cell lines in comparison to control cell lines (Fig. 3a–c, Supplementary Fig. 6a–j, Supplementary Data 3). We also validated the strongest synthetic lethal genetic

interactions with EXO1 loss, with *ZRSR2* and *CDK11B*, in HeLa Kyoto iCas9 cells (Supplementary Fig. 6k–n).

ZRSR2 is one of the top synthetic lethal hits in our genome-wide *EXO1* KO CRISPR screen, and its mutations and deletions represent a significant cancer vulnerability, with frequent driver mutations in

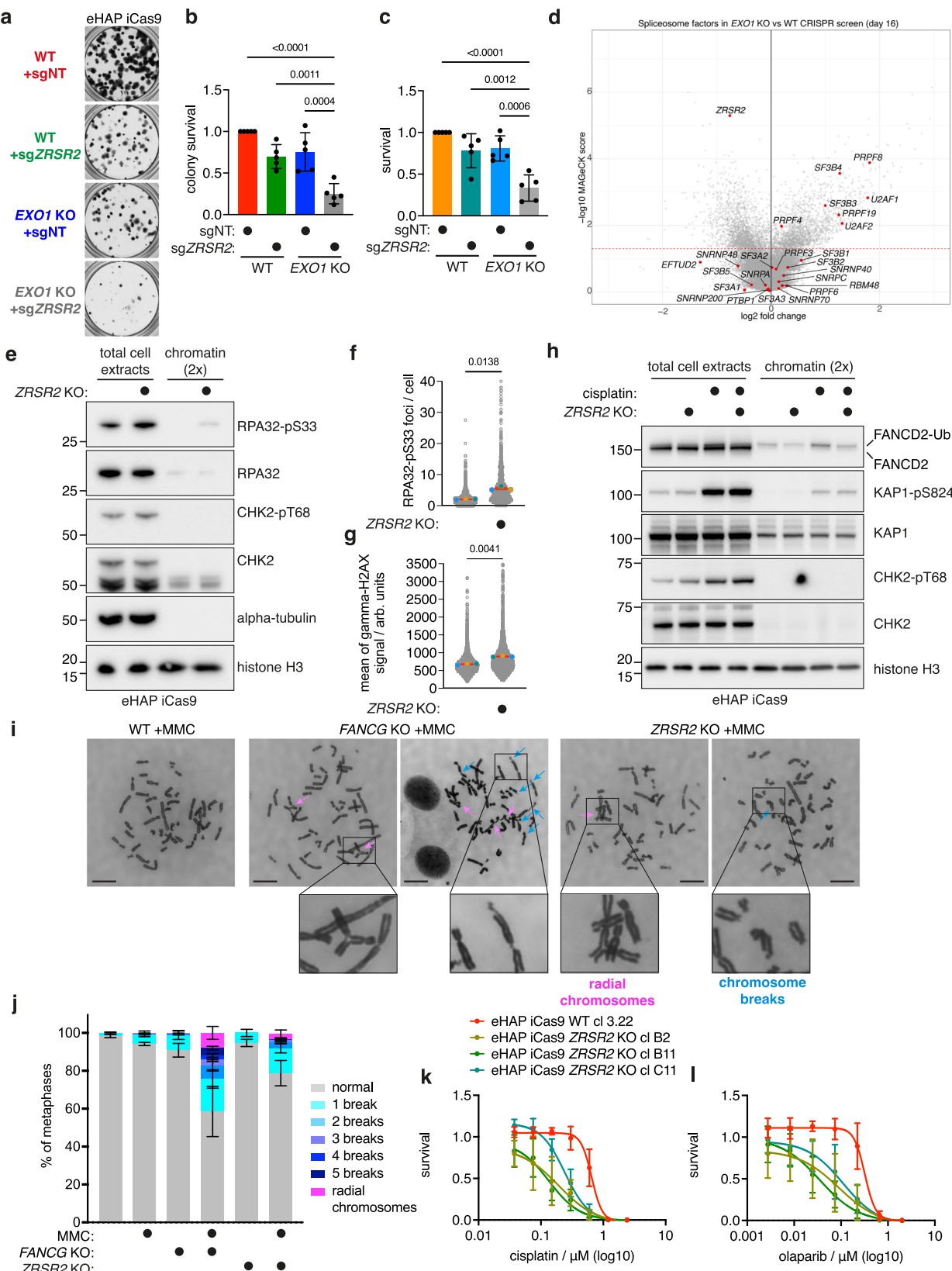

cancers of hematopoietic origin[41]. Interestingly, the *ZRSR2* gene locus is located close to *BRCC3* and *FANCB* on the X chromosome and is homozygously deleted in 6.63% of all cancers through a LINE-1 element retrotransposition event occurring at Xp22.2[30,31]. ZRSR2 has been shown to function as a spliceosome factor regulating distinct steps of U2- and U12-type intron splicing by recognising the 3′ splice site[42]. Notably, loss of ZRSR2 is unique among all spliceosome factors in being synthetic lethal with EXO1 loss (Fig. 3d) and this is unlikely to reflect R-loop accumulation as genes previously implicated in R-loop metabolism were not found as synthetic lethal with EXO1 loss[43] (Supplementary Fig. 7a). Since it has not been previously implicated in the DDR, we investigated the mechanism by which

**Fig. 3 | Loss of the spliceosome factor ZRSR2 confers replication stress, DNA damage and phenocopies FA pathway deficiency. a** Validation of *EXO1* and *ZRSR2* synthetic lethal interaction in eHAP iCas9 cells by clonogenic assay (representative images, *n* = 5, well diameter 15.5 mm). **b** Quantification of clonogenic assay (*n* = 5, mean with SD) in **a** with ordinary one-way ANOVA statistical analysis (*F* = 21.92). **c** Validation of *EXO1* and *ZRSR2* synthetic lethal interaction in eHAP iCas9 cells by CellTitre-Glo viability assay (*n* = 5, mean with SD) with ordinary one-way ANOVA statistical analysis (*F* = 17.84). **d** Volcano plot of the CRISPR screen 'day 16' comparison of *EXO1* KO vs. WT cells with the indicated spliceosome factors. **e** Immunoblotting of total cell extracts and chromatin samples for eHAP iCas9 WT and *ZRSR2* KO. Samples were probed for phosphorylated levels of RPA (pS33), CHK2 (pT68) with controls, as well as alpha-tubulin and histone H3 as a control for equal loading (representative experiment, *n* = 2). **f** Increase of RPA-pS33 foci in *ZRSR2* KO cells in comparison to wild-type cells. Dots in orange, blue and green represent the mean of a biological repeat, with the red line as their mean (mean with SEM, *n* = 3, two-tailed paired *t*-test analysis on the mean of each biological repeat, each mean was calculated on the basis of >711 nuclei). **g** Increase of H2AX-pS139 signal in *ZRSR2* KO cells in comparison to wild type cells. Dots in orange, blue and green represent the mean of a biological repeat, with the red line as their mean (mean with SEM, *n* = 3, two-tailed paired t-test analysis on the mean of each biological repeat, each mean was calculated on the basis of >2060 nuclei). **h** Immunoblotting analysis of total cell extracts and chromatin samples for eHAP iCas9 WT and *ZRSR2* KO without and with cisplatin treatment. Samples were probed for FANCD2, phosphorylated levels of KAP1 (pS824) and CHK2 (pT68) with controls, as well as histone H3 as a control for equal loading (representative experiment, *n* = 3). **i** Representative images of metaphase spreads for eHAP iCas9 'WT + MMC', '*FANCG* KO + MMC' and '*ZRSR2* KO + MMC' (MMC – mitomycin C). Blue arrows point to chromosome breaks, and pink arrows point to radial chromosomes. Scale bars: 10 μm. **j** Quantification of phenotypes observed in metaphase spreads for eHAP iCas9 'WT + MMC', '*FANCG* KO + MMC' and '*ZRSR2* KO + MMC', quantification based on >100 metaphases per repeat and per condition (*n* = 3, mean with SD). **k** Sensitivity of eHAP iCas9 *ZRSR2* KO clones to cisplatin (*n* = 3, mean with SD). **l** Sensitivity of eHAP iCas9 *ZRSR2* KO clones to olaparib (*n* = 3, mean with SD). Source data for **b**, **c**, **e**–**h**, **j**–**l** are provided as a Source Data file.

ZRSR2 inactivation leads to synthetic lethality in combination with EXO1 loss.

To determine if loss of ZRSR2 impacts genome stability, we generated constitutive KOs of *ZRSR2* (Supplementary Fig. 7b) in eHAP iCas9 cells and analysed DDR markers. By immunoblotting we observed enhanced spontaneous phosphorylation of RPA on serine 33, a marker of replication stress, and CHK2 threonine 68 phosphorylation, a DSB signalling marker, indicating that loss of ZRSR2 results in enhanced basal levels of DNA damage (Fig. 3e). We further confirmed this phenotype through imaging of *ZRSR2* KO cells, which showed increased foci for phosphorylated RPA at serine 33 and increased gamma-H2AX intensity (Fig. 3f, g, Supplementary Fig. 7c, d). As ZRSR2 clustered in the CRISPR screen with many factors implicated in the FA pathway, which also exhibit elevated levels of replication stress, we interrogated whether cells deficient for ZRSR2 exhibit features of FA deficiency when challenged with cisplatin or mitomycin C. Cisplatin and mitomycin C induce DNA inter-strand crosslinks, which activate the FA pathway, resulting in the monoubiquitylation of the FANCD2–FANCI complex[44]. Strikingly, we found that monoubiquitylation of FANCD2 is significantly reduced in *ZRSR2* KO cells relative to wild-type controls in response to acute treatment with cisplatin. The monoubiquitylated form of FANCD2 was also reduced on chromatin in *ZRSR2* KO cells (Fig. 3h). Failure to sense and/or repair DNA inter-strand crosslinks in FA-deficient cells is known to result in chromosome aberrations in metaphase spreads. Inducing inter-strand crosslinks with mitomycin C in isogenic wild type, *ZRSR2* KO, and *FANCG* KO cells (Supplementary Fig. 7e–g) resulted in an increase in chromosome breaks in both *FANCG* KO and *ZRSR2* KO cells in comparison to the wild type control (Fig. 3i, j). We also observed an increase in radial chromosomes in both *FANCG* KO and *ZRSR2* KO cells, which is a hallmark of FA-deficient cells (Fig. 3i, j). Furthermore, we demonstrate that like FA-deficient cells, *ZRSR2* KO cells are sensitive to treatment with cisplatin or the PARP inhibitor olaparib (Fig. 3k, l). Finally, we investigated the genetic relationship between the FA pathway and ZRSR2 by examining epistasis in response to EXO1 loss as well as cisplatin. We found that loss of FAAP24 does not confer additional loss of viability of cells upon EXO1 loss in *ZRSR2* KO cells, showing epistasis between ZRSR2 and an FA gene (Supplementary Fig. 7h, i). Additionally, we found that sensitivity of *ZRSR2* KO cells to cisplatin cannot be further exacerbated by depletion of FAAP24, demonstrating epistasis of ZRSR2 and FA pathway for cisplatin sensitivity (Supplementary Fig. 7j). Altogether, these results demonstrate that ZRSR2 deficiency leads to attenuated FA pathway activation, cisplatin sensitivity and genome instability which is epistatic with the FA pathway.

## Mono and combination therapy opportunities for EXO1 inhibitor development

As EXO1 possesses intrinsic catalytic exonuclease activity[45,46], we next explored if this activity is important for the synthetic lethal interactions identified in the CRISPR screen. To this end, we conducted complementation studies in *EXO1* KO cells with either wild-type EXO1 or with an exonuclease catalytic-dead mutant of EXO1 (D173A). We verified expression of *EXO1* variants through immunoblotting and observed that both wild type and catalytic-dead mutant of *EXO1* are expressed to similar levels in *EXO1* KO cell lines with integrated sgRNAs, albeit slightly lower than endogenous levels of EXO1 in eHAP cells (Fig. 4a). Expression of wild type EXO1 in *EXO1* KO cell lines was sufficient to restore viability of both *EXO1-FANCG* and *EXO1-ZRSR2* double KO cell lines to the levels of the inducible KOs of either *FANCG* or *ZRSR2* (Fig. 4b, c). In contrast, expression of the catalytic-dead mutant of EXO1 failed to rescue the viability of either of the double KO cell lines, suggesting that FA- and ZRSR2-deficient cells rely on catalytically active EXO1 and could therefore be targeted through inactivation/inhibition of its catalytic activity (Fig. 4b, c).

To determine if loss of EXO1 can effectively combine with chemotherapies and/or DDR inhibitors to augment the sensitisation of *EXO1-FANCG* and *EXO1-ZRSR2* double KO cell lines, we profiled inducible double KO cells with a panel of DNA-damaging agents and DDR inhibitors. We found that inducible *EXO1-FANCG* double KO cells are strongly sensitised by PARP inhibitors, including olaparib and veliparib, and by cisplatin and camptothecin (Fig. 4d, e, Supplementary Fig. 8a–c). We also found that induction of *ZRSR2* KO in *EXO1* KO cells and treatment of cells with PARP inhibitors further reduced their viability, as did treatment with cisplatin and camptothecin (Fig. 4f, g, Supplementary Fig. 8d–f). These results suggest that FA- and ZRSR2-deficient cancers could be treated with a combination of EXO1 and PARP inhibitors, as well as with a topoisomerase I inhibitor or with platinum-based compounds. We found that double KO *EXO1-FANCG* and *EXO1-ZRSR2* cells are further sensitised by ionising radiation, suggesting that an EXO1-inactivating chemotherapeutic could effectively combine with radiotherapy in the treatment of FA- or ZRSR2-deficient cancers (Fig. 4h, i). These findings were further validated in a HeLa Kyoto iCas9 cell line, suggesting that the observed effects are not dependent on cell line background or cancer type (Supplementary Fig. 8g–j).

## Genetic drivers of EXO1 synthetic lethal interactions

To identify potential drivers of EXO1 synthetic lethality that could inform on the mechanistic basis of the genetic interactions, we performed genome-wide CRISPR rescue screens for both *EXO1-FANCG* and *EXO1-ZRSR2* interactions (Supplementary Fig. 8k, Supplementary

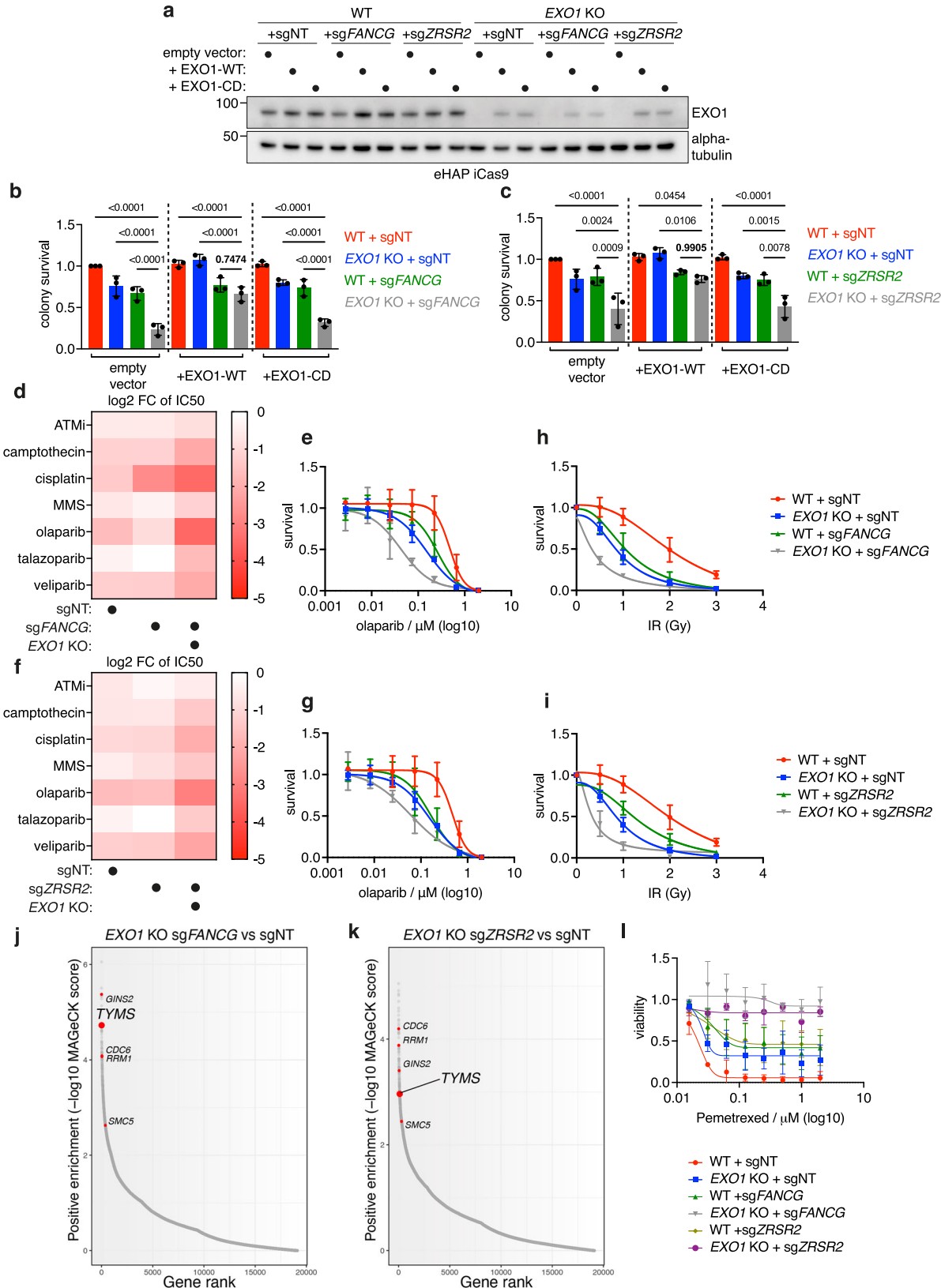

Data 4 and 5). Interestingly, GO Biological Process analysis of rescue genes from both screens revealed an overlap of genes whose loss is better tolerated in the double KO cells relative to *EXO1* KO cells alone, further confirming the similarity of ZRSR2 deficiency to loss of the FA pathway (Supplementary Fig. 8l, m). Notably, *GINS2*, *CDC6*, *SMC5*, *RRM1*, and *TYMS* were identified as rescue genes in both screens, which

are linked to DNA replication and chromosome maintenance. A subset of these genes was previously identified in DNA damage suppressor screens[47] (Fig. 4j, k). We chose to further characterise thymidylate synthase (TYMS), as it is non-essential and can be inhibited pharmacologically with Pemetrexed (TYMS inhibitor), which is currently used as a cancer therapeutic for pleural mesothelioma and non-small cell

**Fig. 4 | Single and combination therapy opportunities for EXO1 inhibitor development. a** Immunoblotting for EXO1 protein levels after complementation with either empty vector, vector constitutively expressing wild type EXO1 (*EXO1-WT*) or vector constitutively expressing catalytic dead mutant of EXO1 (*EXO1-CD*) (alpha-tubulin as control for equal loading). **b** Quantification of clonogenic assay for the complementation experiment for *EXO1* and *FANCG* synthetic lethal interaction (*n* = 3, mean with SD) with ordinary one-way ANOVA statistical analysis (*F* = 41.71). **c** Quantification of clonogenic assay for the complementation experiment for *EXO1* and *ZRSR2* synthetic lethal interaction (*n* = 3, mean with SD) with ordinary one-way ANOVA statistical analysis (*F* = 17.95). **d** Heatmap of log fold change (FC) of IC50 values for survival of '*EXO1* KO + sgNT', 'WT + sg*FANCG*' and '*EXO1* KO + sg*FANCG*' in comparison to 'WT + sgNT' in eHAP iCas9 background after treatment with DNA-damaging agents and inhibitors of DDR factors. IC50 values were calculated from sigmoidal curve fits, and the change was calculated with 'WT + sgNT' value as reference (*n* = 3). **e** Additive sensitivity of eHAP iCas9 double KO cells for *EXO1* and *FANCG* to olaparib (*n* = 3, mean with SD). Example of an experiment that was used to generate the heatmap in (**d**). **f** Heatmap of log fold

change (FC) of IC50 values for survival of '*EXO1* KO + sgNT', 'WT + sg*ZRSR2*' and '*EXO1* KO + sg*ZRSR2*' in comparison to 'WT + sgNT' in eHAP iCas9 background after treatment with DNA-damaging agents and inhibitors of DDR factors; as in (**d**) (*n* = 3). **g** Additive sensitivity of eHAP iCas9 double KO cells for *EXO1* and *ZRSR2* to olaparib (*n* = 3, mean with SD). Example of an experiment that was used to generate the heatmap in (**f**). **h** Additive sensitivity of eHAP iCas9 double KO cells for *EXO1* and *FANCG* to ionising radiation (*n* = 3, mean with SD). **i** Additive sensitivity of eHAP iCas9 double KO cells for *EXO1* and *ZRSR2* to ionising radiation (*n* = 3, mean with SD). **j** Sigmoidal plot for '*EXO1* KO + sg*FANCG* vs *EXO1* KO + sgNT' comparison in CRISPR rescue screen. Data was plotted based on MAGeCK analysis, with highlighted genes of interest. **k** Sigmoidal plot for '*EXO1* KO + sg*ZRSR2* vs. *EXO1* KO + sgNT' comparison in CRISPR rescue screen. Data was plotted based on MAGeCK analysis, with highlighted genes of interest. **l** Rescue of '*EXO1* KO + sg*FANCG*' and '*EXO1* KO + sg*ZRSR2*' sensitivity to TYMS inhibitor Pemetrexed (clonogenic assay) in comparison to 'WT + sgNT', '*EXO1* KO + sgNT', 'WT + sg*FANCG*' and 'WT + sg*ZRSR2*' (*n* = 3, mean with SD). Source data for **a–i, l** are provided as a Source Data file.

lung cancer and is thought to act to slow DNA replication by nucleotide depletion or through accumulation/misincorporation of dUMP in DNA[48,49]. Strikingly, we found that *EXO1-FANCG* and *EXO1-ZRSR2* double KO cells retain viability and tolerate high concentrations of Pemetrexed sufficient to induce death in either wild type cells or single KO cells for *EXO1*, *FANCG* or *ZRSR2* (Fig. 4l). This suggests that enhanced replication stress is a critical driver of the *EXO1-FANCG* and *EXO1-ZRSR2* synthetic lethal interactions, reinforcing our findings of elevated DNA damage and DDR markers in these cells. Evidently, TYMS inhibitor Pemetrexed should not be used in any future EXO1 therapeutic combinations in FA-deficient cancers.

### Deficiency in fork reversal and increase in replication fork speed drive synthetic lethality between EXO1 and FANCG

Since *EXO1-FANCG* double KO exhibits increased replication stress (Fig. 2e, Supplementary Fig. 5f, g), increased phosphorylation of RPA at serine 33 (Fig. 2g, Supplementary Fig. 5i), and tolerance to TYMS inhibition (Fig. 4l), we sought to examine replication fork dynamics in *EXO1-FANCG* double KO. Analysis of replication fork speed from DNA fibres indicated that loss of either EXO1 or FANCG has no significant effect on fork dynamics following 5 days of Cas9 expression induction (Fig. 5a, b). In contrast, we found that *EXO1-FANCG* double KO cells have significantly faster replication forks (Fig. 5a, b). Increased fork speed has been previously observed in cells treated with high concentrations of PARP inhibitors[50]. We therefore compared fork speeds between untreated cells and cells treated with a high concentration of PARP inhibitor olaparib to assess relative fork speeds. High dose of PARPi increased fork speed in all cell lines, with mean fork speed of olaparib-treated wild type and single KO cells comparable to the mean fork speed of *EXO1-FANCG* double KO cells in the unchallenged condition, and with the trend of olaparib-treated *EXO1-FANCG* double KO cells forks exhibiting elevated speeds in comparison to unchallenged cells (Supplementary Fig. 9). This raises the possibility that further increasing fork speeds contributes to the increased cell killing induced by olaparib in *EXO1-FANCG* double KO cells (Supplementary Fig. 9). Since the faster forks observed with high concentrations of PARPi or following co-depletion of RAD51 and geminin[51] have been attributed to a defect in fork reversal, we directly examined fork reversal by electron microscopy (EM) in either unchallenged conditions or upon treatment with HU. In comparison to the unchallenged condition, HU treatment increased fork reversal in wild-type cells and in EXO1- and FANCG-deficient cells, albeit to a lesser extent (Fig. 5c, d). Strikingly, HU treatment failed to induce fork reversal in *EXO1-FANCG* double KO cells (Fig. 5c, d). EM analysis also revealed increased levels of ssDNA behind the forks in *EXO1-FANCG* double KO cells in untreated conditions (Fig. 5e, f), which is consistent with elevated phosphorylated RPA at serine 33 shown above. Release from HU-mediated replication stress

resulted in a further increase of ssDNA in *EXO1-FANCG* double KO cells. We also investigated fork degradation by DNA fibres upon treatment with HU. While loss of FANCG results in fork degradation, which is suppressed by inhibiting MRE11 with mirin, this phenotype is absent in *EXO1-FANCG* double KO cells (Fig. 5g). This suggests that MRE11 and EXO1 drive fork degradation in FANCG-deficient cells (Fig. 5g). Finally, we investigated whether tolerance of *EXO1-FANCG* double KO cells to TYMS inhibition can be explained by a reduction in fork speed[52]. To this end, we analysed replication tracks upon treatment with low dose of Pemetrexed, which had no significant effect on either wild type or cells lacking either EXO1 or FANCG (Fig. 5h). However, low-dose Pemetrexed treatment significantly reduced the speed of replication forks in *EXO1-FANCG* double KO cells (Fig. 5h). These data suggest that tolerance to TYMSi in *EXO1-FANCG* double KO cells is due to restoration of replication fork speeds to wild type levels. We propose that the combination of elevated replication fork speeds resulting from defective fork reversal and accumulation of post-replication ssDNA gaps likely contributes to the synthetic lethality with EXO1 loss in FA-deficient cells (Fig. 6).

## Discussion

In conclusion, this study reveals extensive synthetic lethal genetic interactions with EXO1 in human cells, which defines a framework for EXO1-based targeting of cancers with deficiencies in DDR pathways for which current therapeutic approaches are not sufficiently effective. Our genome-wide CRISPR dropout screen identified established and previously unappreciated DDR factors that participate in mechanisms that are compensated by EXO1. Importantly, we define notable cancer vulnerabilities amongst synthetic lethal hits with EXO1 loss, such as deficiencies in BRCA1-A complex factors, Fanconi Anaemia pathway genes and the splicing factor and tumour suppressor ZRSR2.

ZRSR2 was the only splicing factor that scored as a significant hit in the EXO1 synthetic lethal screen. Since ZRSR2 clustered with other top hits, including many genes previously implicated in FA pathway activation, we explored the possibility that ZRSR2 loss might impact genome stability. Indeed, deletion of ZRSR2 conferred elevated levels of spontaneous phosphorylation of RPA at serine 33, CHK2 on threonine 68 and H2AX on serine 139, indicative of elevated spontaneous replication stress and DNA damage. Consistent with the clustering of ZRSR2 and the FA pathway in the CRISPR screen, we observed that *ZRSR2* KO cells are attenuated for FA pathway activation at the level of FANCD2 monoubiquitylation, which is normally induced upon replication stress. *ZRSR2* KO cells are also sensitive to cisplatin treatment and, upon treatment with MMC, present with radial chromosomes, which are a hallmark of FA pathway deficiencies. Furthermore, ZRSR2 loss was found to be epistatic with the FA pathway with regard to synthetic lethality with EXO1 loss, as well as sensitivity to cisplatin. Finally, we show that the synthetic

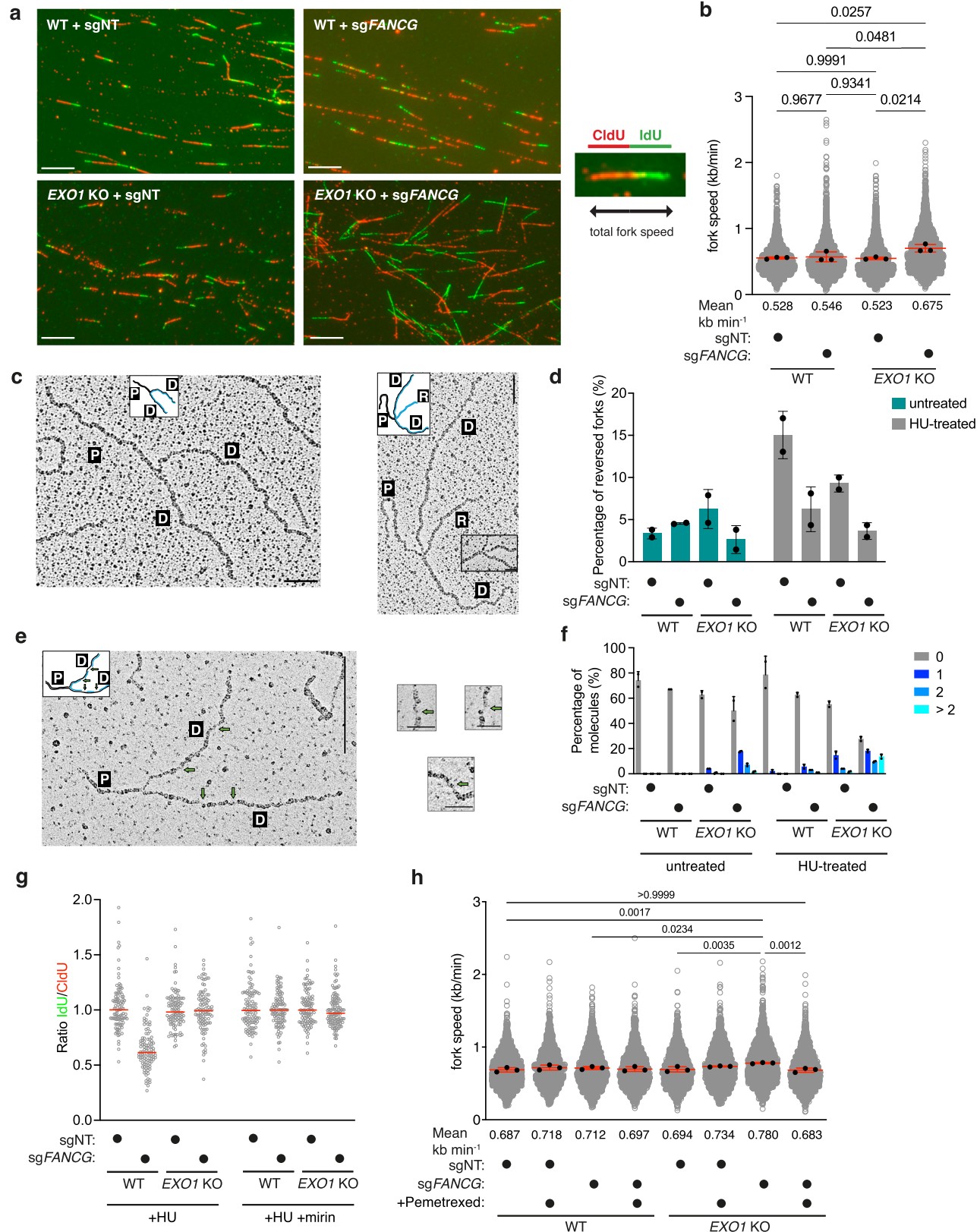

lethality of *EXO1-FANCG* and *EXO1-ZRSR2* double KO cells is suppressed by the inhibition of TYMS, which slows replication fork speeds. We speculate that, unlike other splicing factors, ZRSR2 loss impacts the expression of genes that either directly or indirectly affect the FA pathway, which phenocopies FA pathway deficiency. Precisely how this occurs is uncertain and warrants future investigation.

Mechanistic analysis of the synthetic lethal interaction observed in *EXO1-FANCG* double KO cells revealed evidence of elevated replication stress and DNA damage, modestly increased replication fork speeds, defective replication fork reversal and the accumulation of post-replicative ssDNA gaps (Fig. 6). We further show that PARPi treatment, which also blocks fork reversal, further increases

**Fig. 5 | *EXO1-FANCG* double KO cells are deficient in replication fork reversal, which induces faster replication forks and results in synthetic lethality.**
**a** Representative images of DNA fibres of 'WT + sgNT', 'WT + sg*FANCG*', '*EXO1* KO + sgNT', '*EXO1* KO + sg*FANCG*' after 5 days of Cas9 induction (*n* = 3), with an example of a staining for a DNA fibre (taken from image of 'WT + sgNT') that was measured for total fork speed. Scale bars: 10 μm. **b** Quantification of replication fork speed from the DNA fibres experiment shown in (**a**). Measurements of a minimum of 300 fibres per condition per biological repeat (*n* = 3). Black points represent the mean from each biological repeat, with red lines representing the mean with the standard deviation of those. Means are also noted below the *X*-axis in the numerical value. An ordinary one-way ANOVA statistical analysis was performed on the means of biological repeats (*F* = 6.513). **c** Representative electron micrographs of normal replication fork and reversed replication forks (*n* = 2). Scale bar for large panels: 250 nm, small panels: 50 nm. P: parental strand, D: daughter strand, and R: reversed strand. **d** Quantification of fork reversal in 'WT + sgNT', 'WT + sg*FANCG*', '*EXO1* KO + sgNT', '*EXO1* KO + sg*FANCG*' after 5 days of Cas9 induction, with and without hydroxyurea (HU) treatment following induction; percentage of fork reversal plotted. Error bars are representative of the mean with standard deviation from two independent experiments. **e** Representative electron micrographs of replication fork with extensive ssDNA behind the fork obtained from '*EXO1* KO + sg*FANCG*' cells after 5 days of Cas9 induction (*n* = 2). Scale bar for large panels: 250 nm, small panels: 50 nm. P: parental strand, D: daughter strand. **f** Quantification of ssDNA gaps in 'WT + sgNT', 'WT + sg*FANCG*', '*EXO1* KO + sgNT', '*EXO1* KO + sg*FANCG*' after 5 days of Cas9 induction, with and without HU treatment following induction. Gaps observed per fork are quantified as 0, 1, 2 and more than 2. Error bars are representative of the mean with standard deviation from two independent experiments. **g** Quantification of fork degradation rate (IdU to CldU ratio) via DNA fibres experiment following 4 mM HU treatment without and with mirin (representative experiment from three independent experiments, red lines represent median values). Measurements of a minimum of 100 fibres per condition per biological repeat. **h** Quantification of replication fork speed from DNA fibre experiment without and with TYMS inhibitor Pemetrexed. Measurements of a minimum of 300 fibres per condition per biological repeat (*n* = 3). Black points represent the mean from each biological repeat, with red lines representing the mean with the standard deviation of those. Means are also noted below the *X*-axis in the numerical value. Two-way ANOVA statistical analysis was performed on the means of biological repeats. Source data for **b**, **d**, **f** and **h** are provided as a Source Data file.

replication fork speed, whereas slowing replication forks through TYMS inhibition fully suppresses this phenotype. Finally, we show that *FANCG* KO cells exhibit increased fork degradation, which is MRE11- and EXO1-dependent. These data raise the possibility that the synthetic lethality observed in *EXO1-FANCG* double KO cells arises from elevated fork speeds and exposure of ssDNA, which is a consequence of defects in fork reversal. FANCG has been previously implicated in the repair of DSBs in chicken DT40 cells through homologous recombination[53]. Furthermore, the loss of EXO1 has been shown to be synthetically lethal with BRCA1 deficiency due to the loss of single-stranded annealing (SSA)[33]. It is plausible that the exposure of ssDNA resulting from the lack of fork reversal in *EXO1-FANCG* double knockout cells leads to DSBs (Fig. 2h), necessitating repair through HR or SSA-mediated mechanisms. The absence of FANCG and EXO1-mediated HR and SSA in these cells could result in cellular lethality, with the damage originating from the roles of FANCG and EXO1 in maintaining replication fork stability. Further investigation is required to elucidate the precise mechanisms underlying this phenomenon.

Importantly, homozygous deletions of factors identified as synthetic lethal with EXO1 loss in this study are found in 8.49% of all cancers (Table 1), defining a significant proportion of cancers potentially targetable by EXO1 inactivation, which is in addition to the previously identified population of BRCA1-deficient cancers[33]. The population of cancers targetable by EXO1 inactivation is likely an underestimation, as compound heterozygosity of loss-of-function mutations is observed in the majority of the 6.58% of all cancers mutated for FA core complex and FANCM complex factors (Table 1), which could be further increased through epigenetic silencing. As these deficiencies in cancers are not currently associated with an effective therapeutic strategy, these findings bring EXO1 to the forefront of targets for their treatment. Altogether, our data argue for the development of EXO1 inhibitors, which would be differentiated from and combine effectively with existing therapies to expand the therapeutic options for treating DDR-deficient cancers.

## Methods

### Cell lines
eHAP is a diploidised HAP1 cell line, derived from the chronic myelogenous leukaemia KBM-7 cell line (haploid originally purchased from Horizon Discovery). eHAP iCas9[20] cell lines were grown in IMDM media with 10% Tet-free FBS and 1% penicillin/streptomycin. HeLa Kyoto iCas9[54] cell lines were grown in DMEM media with 10% Tet-free FBS and 1% penicillin/streptomycin. PL11 cell line (obtained from Alan D'Andrea, Dana-Farber Cancer Institute, Harvard Medical School) was grown in RPMI 1640 media with 15% FBS and 1% penicillin/streptomycin. HEK293FT cell line (originally purchased from Invitrogen) was grown in either DMEM media or IMDM media with 10% FBS and 1% penicillin/streptomycin. All cell lines were grown at 37 °C in 5% $CO_2$. The diploidised eHAP cell line was karyotyped by low-pass sequencing at The Francis Crick Institute.

### Plasmids and cloning
For the generation of KO cell lines, lenti-sgRNA plasmids (-Puro and -Hygro, Addgene #104990 and #104991)[55] or lentiCRISPR v2-Hygro (Addgene #91977)[56] were used for cloning of sgRNA sequences (Supplementary Data 6) for targeting the gene of interest (GOI) using the GeCKO protocol[57]. For the generation of expression vectors for complementation experiments, the piggyBac transposon delivery system was used with a transposase expression vector and piggyBac-EF1α-IRES-Neo expression vector (adapted for Gateway cloning technology, Invitrogen). EXO1 wild type or catalytic dead (D173A) constructs were amplified from Addgene plasmids #111621[13] and #111622[13] using oligos attB1-EXO1-FW and attB2-EXO1-REV (Supplementary Data 7) with Q5 High-Fidelity DNA Polymerase (NEB) to insert them into pDONR221 vector through BR reaction. FANCC wild-type ORF was amplified from a cDNA library (generated with SuperScript III cDNA kit, Invitrogen) using oligos attB1-FANCC-FW and attB2-FANCC-REV (Supplementary Data 7). LR reaction was used to insert the constructs into piggyBac-EF1α-IRES-Neo vector. *E.coli* Stbl3 competent cells (Invitrogen, C737303) were used for propagation of lentiviral vectors, *E.coli ccd*B competent cells (Invitrogen, A10460) were used for propagation of parental Gateway vectors, while *E.coli* DH5α competent cells (Thermo Scientific, EC0112) were used for all other vectors. Bacteria were grown using standard LB media at either 37 °C or at 30 °C for lentiviral vectors, with antibiotics ampicillin or kanamycin (at 100 μg/ml final concentration) for selection.

### Generation of lentiviruses and transduction of cell lines
Lentiviruses were produced in HEK293FT cell line by transfection with plasmids expressing capside proteins (pLP1, pLP2 and VSVG) and with lentiviral expression plasmid as described previously[20]. Briefly, $4 \times 10^6$ HEK293FT cells were seeded in a 10 cm dish for transfection with plasmids (2.83 μg pLP1, 1.33 μg pLP2, 1.84 μg VSVG, 5 μg expression plasmid) using 20 μl Lipofectamine 2000 (Thermo Fisher Scientific) according to the manufacturer's instructions, and with media change ~16 h after transfection. Supernatant with lentiviruses was collected ~72 h after transfection and cleared from cells through a 0.45 μm filter. Prior to transductions, the media on cells was changed to contain

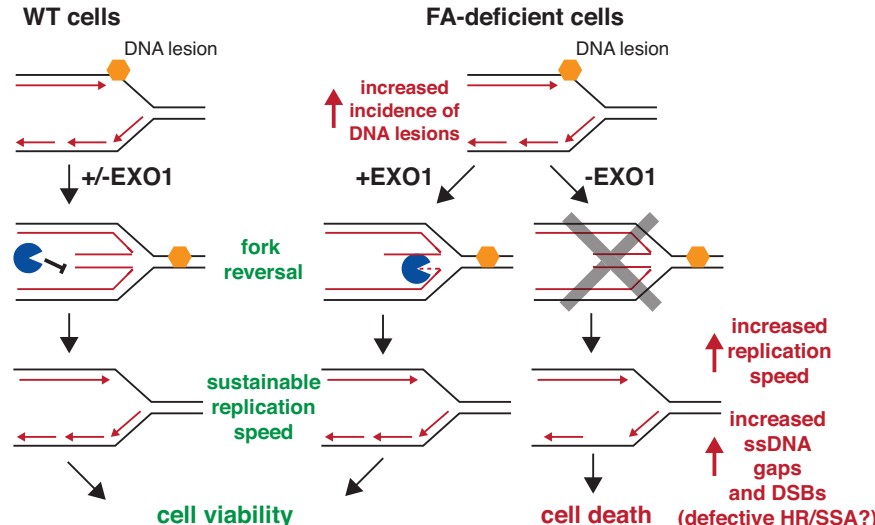

**Fig. 6 | Model for mechanistic causes of synthetic lethality with loss of EXO1 in FA-deficient cells.** In cells with a functional Fanconi Anaemia pathway, replication forks that encounter a DNA lesion can undergo fork reversal as a way of safe-guarding genome integrity, ensuring sustainable replication fork speed and cell viability. However, in FA-deficient cells, in which genome stability is challenged with a higher incidence of DNA lesions, this mechanism of genome stability maintenance critically relies on EXO1. Loss of EXO1 in FA-deficient cells leads to a defect in replication fork reversal, elevated replication fork speed and an increase in post-replicative ssDNA gap formation, which ultimately contributes to DNA damage and results in cell death.

polybrene (Sigma) at a final concentration of 0.8 µg/ml. When required, transduced cells were spun at 290×g for 90 min at 37 °C.

### Generation of constitutive KO cell lines with CRISPR-Cas9
Constitutive knockout cell lines (for *EXO1*, *FANCG* or *ZRSR2*) were generated by induction of Cas9 with doxycycline (1 µg/ml final concentration) a day prior to transfection with a lenti-sgRNA-Puro with sgRNA sequence targeting GOI. 1 µg of plasmid DNA was used to transfect $2 \times 10^5$ cells with Lipofectamine 2000 (Invitrogen). Transfected cells were selected with puromycin, after which they were seeded in 96-well plates at 0.5 cell/well density for single KO clone derivation (efficiency of the editing in the pool was checked by immunoblotting or sequencing). Clones were tested by sequencing (and immunoblotting for EXO1 and FANCG) to confirm knockout of GOI and additionally tested for puromycin sensitivity to ensure that the vector used to transfect the cells had not integrated into the cellular genome.

### Generation of complemented cell lines with piggyBac transposon system
For the generation of complemented cell lines, $2 \times 10^5$ cells were transfected with 1 µg of the respective piggyBac expression vector together with 0.7 µg of transposase expression vector using Lipofectamine 2000. Cells with integrated piggyBac vectors were selected with 0.4 µg/ml G418 sulfate (Geneticin, Gibco) over the course of 8 days (passaging every two days). Expression of EXO1 wild type or mutant was checked by immunoblotting.

### Cas9 editing efficiency assay
Cells were transduced with lentiviruses expressing BFP and GFP, as well as sgRNA targeting GFP sequence (Addgene #67980)[58]. The following day, transduced cell lines were split without and with doxycycline (1 µg/ml final concentration). Cells were passaged until 96 h post transduction, when editing (on 20000 BFP+ cells) was analysed by flow cytometry with BD LSRFortessa Cell Analyzer and using BD FACSDiva software. Gating was performed manually with FlowJo software, where the first BFP+ population was gated (450-50-A vs. FSC-A) and after which the BFP+/GFP− population was gated using a non-doxycycline control (450-50-A vs. 530-30-A).

### Generation of inducible KO cell lines with CRISPR-Cas9
Cells with inducible Cas9 were transduced with lentiviruses containing lenti-sgRNA-Puro (or -Hygro) with sgRNA targeting GOI. Transduced cells were selected with puromycin (0.4 µg/ml for eHAP iCas9 or 0.5 µg/ml for HeLa Kyoto iCas9 cell lines) for 48–72 h before expansion and testing for KO.

### Sequencing of inducible cell lines and deconvolution of sequencing chromatograms
To analyse editing efficiency for the targeted locus, samples were collected at seeding for a particular assay. PureLink Genomic DNA Mini Kit (Invitrogen) was used to obtain DNA for PCR (with Q5 High-Fidelity DNA Polymerase, NEB), followed by PCR purification and sequencing (both Genewiz services). Chromatograms were analysed through Synthego ICE Analysis tool and Decodr[59] to deconvolve traces and determine the proportion of sequences with frameshift deletions.

### CRISPR-Cas9 screening and analysis
Wild-type and *EXO1* KO cell lines were transduced with Brunello library[60] from Addgene containing sgRNAs targeting 19,114 genes, with 4 sgRNAs per gene, and with 1000 non-targeting sgRNAs. Transductions (performed in a biological triplicate on cells that were passaged separately through expansion) were done on $100 \times 10^6$ cells (in 12-well plates) at MOI of 0.4 for a ×500 representation of each sgRNA (titre required for both cell lines was determined prior to experiment). Cells with viruses were spun at 290×g for 90 min at 37 °C. The following day, cells were trypsinised to transfer into 5-layer flasks (Falcon) and puromycin was added for selection. $40 \times 10^6$ cells for each replicate were passaged onwards every two days to ensure that sgRNA representation is maintained. $60 \times 10^6$ cells of transduced cells were harvested for freezing in 1 ml of Recovery Cell Culture Freezing Medium (Gibco) for storage at −80 °C. Replicates were then treated with doxycycline (1 µg/ml final concentration) for 6 days. Samples were collected at day 6 and day 16 for library preparation and sequencing (harvesting $60 \times 10^6$ cells per sample). Genomic DNA was extracted with the PureLink Genomic DNA Mini Kit per manufacturer's instructions (Invitrogen; 16 columns per sample). 200 µg gDNA was used for PCRs with Ex Taq polymerase (TaKaRa) to prepare sgRNA libraries following Broad Institute's protocol[57,60] with P5 mix of oligos and barcoded P7 oligos

(Supplementary Data 8). Libraries (354 nt product) were gel purified with QIAquick Gel Extraction Kit (QIAGEN), followed by MinElute PCR Purification kit (QIAGEN) for buffer exchange, and analysed with Agilent 2100 Bioanalyzer. Libraries were sequenced with HiSeq 4000 (Illumina) in 100 bp single-end configuration for 30 million reads per sample. MAGeCK analysis was done as described before[20]. Briefly, trimming of raw data was done by obtaining 20 bp after the first occurrence of "CACCG" in the read sequence. Following that, trimmed reads were mapped with BWA (version 0.5.9-r16) to a database of guide sequences for the human CRISPR Brunello lentiviral pooled library (downloaded from Addgene: https://www.addgene.org/pooled-library/broadgpp-human-knockout-brunello/) using parameters "-l 20 -k 2 -n 2". sgRNA counts were obtained after filtering the mapped reads for those that had zero mismatches, and mapped to the forward strand of the guide sequence. The MAGeCK 'test' command (version 0.5.7) was used to perform the sgRNA ranking analysis between the relevant conditions with parameters "–norm-method total–remove-zero both."

A Principal Component Analysis (PCA) of raw reads for each experimental condition in the CRISPR-Cas9 dropout screens was generated using the prcomp function in R with arguments centre = TRUE and scale = TRUE. The resulting matrix was visualised by plotting PC2 values against PC1 values for each experimental condition.

For rescue screens eHAP iCas9 *EXO1* KO cells, which were previously transduced with Brunello library in triplicate for the dropout screen and frozen, were recovered and transduced with lenti-sgRNA-Hygro with sgRNAs for either *FANCG* or *ZRSR2* or non-targeting control (at MOI of 1 based on predetermined titre). Transduced cells were selected with hygromycin, and Cas9 expression was induced with doxycycline. Cells were passaged every 2–3 days (keeping $40 \times 10^6$ cells representation) and samples were collected 10 days after Cas9 expression induction. Libraries were prepared for sequencing as described above and sequenced with NovaSeq 6000 (Illumina) in 100 bp paired-end configuration for 30 million reads per sample. Raw data were trimmed, aligned to sgRNA sequences from the Brunello library, and counted using the built-in count function of MAGeCK[61]. The MAGeCK test command was used to compare sgRNA counts between the indicated experimental conditions and to generate sgRNA ranking analyses.

Plots were generated using the R programming language data visualisation package ggplot2.

### Gene Ontology (GO) Biological Process pathway analysis
Analyses were performed through Enrichr portal[62–64] where lists of genes were analysed through GO Biological Process library (2023 version used for analysis shown in study). Calculations for enrichment of identified categories and pathways were extracted from data (*p*-values calculated by Fisher's exact test) and as a −log value plotted on bar graphs for the top categories.

### Clonogenic assay for viability
Cells were seeded in 24-well plates at 200 cells/well a density (in 4 technical replicates) and colonies were allowed to grow for 6 days before fixing with 0.5% crystal violet stain with 20% methanol. Stain was rinsed with water and after drying plates were imaged with Gel-Count (Oxford Optronics). GelCount software was used to analyse colony numbers in each well. Colony count was averaged over technical replicates and normalised against the control.

### Cell Titre Glo assay for viability
Cells were seeded in opaque 96-well plates at a 200 cells/well density (in 3 technical replicates) and allowed to grow for 6 days before cell lysis with CellTitre-Glo reagent (Promega). Luminescence signal was readout using CLARIOstar Plus Microplate Reader (BMG Labtech) and exported via its MARS Data Analysis software. Values for the signal in technical replicates were averaged before normalising against the control.

### Sensitivity to DNA-damaging drugs and DDR factor inhibitors
Cells were seeded in opaque 96-well plates at 200 cells/well density (for eHAP cells) or at 500 cells/well density (for HeLa Kyoto cells). Compounds were added to the media the next day and cells were allowed to grow for 5 more days before cell lysis with CellTitre-Glo reagent and processing as above. Luminescence signal values were normalised against untreated control wells for each cell line. Compounds purchased from Selleckchem used in this study: PARPi olaparib (AZD2281), PARPi talazoparib (BMN 673), PARPi veliparib (ABT-888), ATMi (KU-55933), ATRi (Ceralasertib, AZD6738), TYMSi (Pemetrexed, LY231514). Inhibitor against FEN1[65] was a kind gift from Artios Pharma Ltd. Agents purchased from Sigma-Aldrich used in this study: cisplatin (CDDP), camptothecin, methyl methanesulfonate, hydroxyurea, potassium bromate, and mitomycin C. Methanol-free formaldehyde (Pierce) was purchased from Thermo Scientific.

### Sensitivity to ionising radiation
Cells were seeded 4 h before irradiation for a clonogenic assay for treatments with the indicated dose. For biological triplicate cells were seeded and irradiated separately. Irradiations were carried out using the XStrahl RS320 irradiation cabinet. The dose rate was determined by probe reading, and exposure times were adjusted based on this prior to all irradiations. The treatments were carried out at 300 kV, 10 mA with a dose rate of -0.5 mGy/min.

### Cell proliferation assay with Incucyte
For proliferation assay of eHAP cell lines, cells were seeded 4 days after Cas9 expression induction with doxycycline into an opaque 96-well plate (black wells), with 1000 cells seeded per well (seeding in three wells for each cell line). For proliferation assay of PL11 cell lines, cells were seeded 5 days after transduction with lentiCRISPR-sgNT or sg*EXO1*, with 4000 cells seeded per well (seeding in six wells for each cell line). Cell proliferation was tracked using Incucyte ZOOM System (Sartorius) through phase confluency measurement by imaging every 3 h until indicated (taking 4 images per well). Software Incucyte Basic Analyzer (Sartorius) was used to obtain information on the percentage of confluency, which, averaged for technical replicates, was directly plotted against time for each of the cell lines. For statistical analyses, proliferation profiles were analysed against the best fit and slopes were compared within conditions.

### Apoptosis assay with Incucyte
Media on cells seeded for the assay (coupled to cell proliferation assay) was changed to contain NucView 488 (BioTracker) apoptotic-induced nuclear dye (at 1 mM final concentration). Cells were placed in the Incucyte ZOOM System (Sartorius) to follow cell proliferation (through phase confluency measurement) and apoptosis (through the green channel to identify nuclei of apoptotic cells). Software Incucyte Basic Analyzer (Sartorius) was used to obtain information on the percentage of confluency and the number of green nuclei. To be able to correlate confluency to number of nuclei for the ratios analysis, eHAP iCas9 cell line was separately seeded in an opaque 96 well plate in a range of 1000–16,000 cells per well (in a technical duplicate) and DRAQ5 dye (at 5 µM final concentration) was added with media the following day to generate a dataset of DRAQ5 images and brightfield images. The paired dataset was then used to train a UNet-based model[66], which was then applied to generate predicted DRAQ5 images for the brightfield images from the experimental data. Cellpose[67] was used to segment the nuclei from the predicted DRAQ5 images and to derive the total nuclei counts from the number of segmented nuclei. For the images captured on the green channel, Cellpose segmentation was run directly without the prediction step to generate segmentations and

counts. Ratio of apoptotic nuclei vs. total nuclei was calculated for each cell line and timepoint and plotted against time for each of the biological replicates.

### Cell extracts

To make total cell extracts, cells were counted after harvesting (typically $2 \times 10^5$ cells from each sample would be taken for analysis). Cells were pelleted for 5 min at $2300 \times g$ and washed once with PBS. Pellets were resuspended in a modified Laemmli buffer (50 mM Tris–HCl pH 6.8, 100 mM DTT, 2% SDS, 0.1% bromophenol blue, 10% glycerol) to make a total cell lysate (50 µl of buffer would be used for lysis of $2 \times 10^5$ cells). Chromatin digestion was achieved by Benzonase (Millipore) treatment (with samples rotating for 30 min at room temperature) using a ratio of 25 U per $2 \times 10^5$ cells. Samples were then boiled for 10 min at 95 °C.

### Chromatin fractionation

Cells were counted after harvesting and separated for total cell extracts and for chromatin fractions (typically $10^6$ cells for total cell extracts, with 150 µl of modified Laemmli buffer to resuspend the cell pellets for total cell extracts, and typically $2 \times 10^6$ cells for the final chromatin pellet to be resuspended in 150 µl of modified Laemmli buffer to make '2× chromatin' samples). Pelleted cells were washed in PBS after which they were resuspended in the CSK buffer (10 mM PIPES/KOH at pH 6.8, 100 mM NaCl, 300 mM sucrose, 1 mM EGTA, 1 mM MgCl$_2$, with freshly added 1 mM DTT together with phosphatase (phosSTOP, Roche) and protease inhibitors cocktails (cOmplete, Roche); typically, 1 ml of buffer was used for $2 \times 10^6$ cells) and incubated on ice for 30 min. Chromatin was pelleted at $300 \times g$ for 3 min at 4 °C and washed with 1 ml of CSK buffer. Supernatant was removed, and a modified Laemmli buffer was added together with Benzonase (500U per chromatin pellet obtained from $2 \times 10^6$ cells). Chromatin was then digested for 30 min with shaking at room temperature and boiled for 10 min at 95 °C.

### Immunoblotting

Samples were resolved using NuPAGE SDS–PAGE system (Invitrogen, 4-12% Bis–Tris, 12% Bis–Tris or 3–8% Tris–Acetate gels) and transferred onto nitrocellulose (Amersham Protran 0.2 µm) or PVDF membranes (Amersham Hybond P 0.45 µm). Membranes were then blocked in 5% non-fat milk in TBST buffer (or in 3% bovine serum albumin in TBST buffer for antibodies against phosphorylated proteins), followed by primary antibody incubation and secondary antibody incubation. Primary antibodies used were: anti-EXO1 (ab95068, Abcam, rabbit, 1:1000), anti-alpha-tubulin (clone B-5-1-2, T6074, Sigma, mouse, 1:10000), anti-vinculin (clone hVIN-1, ab11194, Abcam, mouse, 1:20000), anti-FANCG (clone F-8, sc-393382, Santa Cruz, mouse, 1:1000), anti-KAP1-pS824 (ab70369, Abcam, rabbit, 1:1000), anti-KAP1 (clone 20C1, ab22553, Abcam, mouse, 1:1000), anti-p53-pS15 (9284S, CST, rabbit, 1:1000), anti-p53 (clone 1C12, 2524S, CST, mouse, 1:1000), anti-CHK2-pT68 (2661S, CST, rabbit, 1:1000), anti-CHK2 (clone 7, 05-649, Millipore, mouse, 1:500), anti-H2AX-pS139 (gamma-H2AX, clone JBW301, 05-636, Millipore, mouse, 1:1000), anti-histone H3 (ab1791, Abcam, rabbit, 1:5000), anti-RPA-pS33 (A300-246A, Bethyl Laboratories, rabbit, 1:1000), anti-RPA (clone 9H8, ab2175, Abcam, mouse, 1:1000), anti-FANCD2 (clone EPR2302, ab108928, Abcam, rabbit, 1:1000), anti-RMI2 (ab122685, Abcam, rabbit, 1:500), anti-FAM175A/Abraxas1 (clone EPR6310(2), ab139191, Abcam, rabbit, 1:1000), anti-BRCC36 (A302-517A-M, Bethyl Laboratories, rabbit, 1:1000), anti-FANCC (clone 8F3, MABC524, Sigma, mouse, 1:1000). Anti-FAAP24 antibody (rabbit, 1:1000) was a kind gift from Stephen West[68] and anti-APITD1/MHF1 antibody (rabbit, 1:1000) was a kind gift from Weidong Wang[69]. Secondary antibodies used were goat anti-mouse immunoglobulins/HRP (Dako) and swine anti-rabbit immunoglobulins/HRP (Dako) (used 1:5000). Images of chemiluminescence signals were

obtained on Bio-Rad ChemiDoc MP after membrane incubation with either Clarity Western ECL or Clarity Max Western ECL (both Bio-Rad).

### Cell cycle analysis

For analysis of cell cycle phases, cells were incubated for 15 min with EdU (at a final concentration of 10 µM in calibrated media for temperature and CO$_2$ content) and fixed in 4% PFA in PBS. Fixed cells were permeabilised with 0.5% Triton X-100 in PBS and washed in 1% BSA in PBS prior to click reaction to Alexa Fluor 488 with Click-iT EdU Flow Cytometry Assay kit (Thermo Fisher Scientific). DNA was stained with DAPI at 2 µg/ml, a final concentration. Signals for EdU and DAPI for each cell (total of 10,000 single cells per sample) were acquired through flow cytometry with BD LSRFortessa Cell Analyzer and using BD FACSDiva software. Gating for cell cycle analysis was performed manually with FlowJo software. Debris was removed using Forward Scatter Area (FSC-A) vs Side Scatter Area (SSC-A). 'Single cells' were then gated from doublets/clumps using the FSC-H vs. FSC-A ratio and the gating of objects with FSC-A:FSC-H ratio of ~1.

### Immunofluorescence and high content imaging

Cells were seeded at either 5000 cells/well for wild type cells or 10,000 cells/well for KOs in opaque Perkin Elmer 96-well plates for next day fixation with 4% PFA in PBS (20 min at room temperature). Fixed cells were washed with PBS and kept in PBS at 4 °C until antibody incubation. For immunofluorescence staining, fixed cells were incubated in ADB (0.1% Triton X-100, 0.1% saponin, 10% goat serum (Sigma, G6767) in PBS) for blocking, after which they were incubated in the same buffer with primary and then secondary antibodies, as well as washes in between incubations. Primary antibodies used were anti-RPA-pS33 (A300-246A, Bethyl Laboratories, 1:5000) and anti-gamma-H2AX (clone JBW301, 05-636, Millipore, 1:5000). Secondary antibodies used were goat anti-rabbit Alexa Fluor 594 (A11037, Invitrogen, 1:1000) and goat anti-mouse Alexa Fluor 488 (A11029, Invitrogen, 1:1000). Cells were washed in PBS three times after secondary antibody incubation, with second wash containing DAPI stain at 1 µg/ml final concentration. Stained cells were kept in PBS at 4 °C until imaging, which was performed using Opera Phenix High Content Screening System (Perkin Elmer) with ×40 water objective (NA 1.1, confocal), following analysis with Harmony software (Perkin Elmer).

### Metaphase chromosome spreading analysis

Upon treatment with 150 nM mitomycin C for 24 h, cells were arrested in metaphase with nocodazole treatment (at a final concentration 100 ng/ml) for 4 h. Metaphase cells were harvested, fixed and processed for chromosome spreading as described before[70]. Briefly, metaphase cells were swelled in a hypotonic solution (medium:deionized water at 1:2 ratio) for 6 min at room temperature. After pelleting, cells were fixed with freshly made Carnoy's buffer for 15 min at room temperature and spun down. The fixation step was repeated four times, and samples were kept at −20 °C until ready to be dropped on slides. Suspension of cells in Carnoy's buffer in ~100 µl was dropped on a clean slide from about 80 cm distance and dried at room temperature. Slides were incubated with 3% Giemsa stain in PBS for 6 min at room temperature. After drying, slides were mounted with DPX mountant (Sigma). Spreads were imaged using a Nikon Ti2 microscope with Prime 95B camera (Photometrics), Plan Apochromat ×100/1.45 NA Oil objective lens, controlled by Nikon NIS-Elements. Each metaphase was scored for radial chromosome presence and for chromosome breaks (categorised for the number of breaks per metaphase).

### DNA fibre analysis

Cells for DNA fibre analysis were seeded at a density of $6 \times 10^5$ cells/6 cm dish for labelling of DNA the following day. Where indicated, prior to labelling, cells were pretreated with PARP inhibitor olaparib (4 h incubation with 10 µM olaparib) or TYMS inhibitor Pemetrexed

(24 h incubation with 100 nM Pemetrexed). Nucleotide analogues CldU and IdU were dissolved at a final concentration of 25 and 250 µM, respectively, in temperature- and $CO_2$-equilibrated media before sequential incubation pulses of 20 min each, before ice-cold PBS washes and harvesting for fibre spreading. DNA was spread on microscope slides at ~1000 cells/slide density with spreading buffer (200 mM Tris–HCl pH 7.4, 50 mM EDTA, 0.5% (w/v) SDS), fixed with methanol:acetic acid buffer (3:1) and air dried. Staining of slides was performed as follows: slides were washed in water twice, rinsed with 2.5 M HCl and then denatured with 2.5 M HCl for 75 min. After rinsing in PBS, slides were washed with blocking solution (1% BSA and 0.1% Tween-20 (v/v) in PBS), after which they were incubated in blocking solution for 1 hour. Rat anti-BrdU (ab6326, Abcam) was used for immunolabelling of CldU, while mouse anti-BrdU (clone B44, 347580, Becton Dickinson) was used for immunolabelling of IdU. Primary antibodies in blocking solution were incubated on slides for 1 h, followed by three rinses in PBS and three washes in blocking solution. Anti-rat AlexaFluor 594 and anti-mouse AlexaFluor 488 were used as secondary antibodies for immunolabelling for 90 min, followed by two rinses in PBS, three washes in blocking solution and two final rinses in PBS before adding Prolong Gold mounting media with coverslip. Imaging of stained DNA fibres was performed using a Nikon Ti2 microscope with Prime 95B camera (Photometrics), Plan Apochromat ×100/1.45 NA Oil objective lens, controlled by Nikon NIS-Elements. ImageJ software was used for image processing and analysis. Fork protection DNA fibre analysis was performed as described previously[71]. Briefly, after sequential pulse-labelling with 30 µM CldU and 250 µM IdU for 20 min, cells were treated with 4 mM hydroxyurea for 3 h. Cells were then collected and resuspended in PBS at $2.5 \times 10^5$ cells/ml. Labelled cells were then mixed at a 1:1 (v/v) ratio with unlabelled cells before adding 2.5 µl of cells to 7.5 µl of lysis buffer (200 mM Tris–HCl pH 7.5, 50 mM EDTA, 0.5% (w/v) SDS) on a glass slide. Following 8 min incubation, slides were tilted at an angle (15°–45°) to spread DNA fibres. Slides were then air dried and fixed in methanol:acetic acid buffer (3:1) at 4 °C overnight. For staining, DNA fibres were denatured with 2.5 M HCl for 1 hour, washed with PBS and blocked with 0.2% Tween 20 (v/v) in 1% BSA/PBS for 40 min. Immunolabelling of CldU and IdU tracks was performed with anti-BrdU antibodies as described above (for 2.5 h in the dark, at room temperature), followed by 1 h incubation with secondary antibodies in the dark at room temperature, using anti-mouse Alexa Fluor 488 (A11001, Invitrogen, 1:300) and anti-rat Cy3 (712-166-153, Jackson Immuno-Research Laboratories, Inc., 1:150). Fibres were visualised and imaged by Carl Zeiss Axio Imager D2 microscope using ×63 Plan Apo 1.4 NA oil immersion objective. Data analysis was carried out with ImageJ software. Subtle variance between fork speed experiments could be due to assays being performed by two different researchers over an 18 month period with different numbers of samples (experiments in Fig. 5b in comparison with experiments in Fig. 5h and Supplementary Fig. 9). Variance could also result from differences in batches of reagents used, differences in media composition and modest changes in atmospheric oxygen conditions. Despite variance in absolute numbers between several experiments, statistical significance was observed, confirming the robustness of the findings.

### Electron microscopy analysis of replication forks
Replication fork architecture was analysed through standard protocol[71]. Briefly, the asynchronous population of the cultured cells was treated with 4 mM HU for 3 h where indicated. Fork architecture was in vivo cross-linked by 10 µg/ml of 4,5′,8-trimethylpsoralen and pulse irradiation with 365 nm monochromatic UV light (BLX312 UV cross-linker, Vilber Lourmat). DNA was extracted with lysis buffer (1.28 M sucrose, 40 mM Tris–HCl pH 7.5, 20 mM $MgCl_2$, and 4% Triton X-100) and digested in digestion buffer (800 mM guanidine–HCl, 30 mM Tris–HCl pH 8.0, 30 mM EDTA pH 8.0, 5% Tween-20, and 0.5%

Triton X-100) in the presence of 1 mg/ml Proteinase K at 50 °C for 2 h. DNA was then collected by phase separation using chloroform:isoamyl alcohol (24:1) and precipitated with 0.7 volume of isopropanol. DNA was then washed with 70% ethanol before drying and resuspending in TE buffer. 12 µg of the genomic DNA was then digested with PvuII-HF (New England Biolabs) restriction enzyme. Digested genomic DNA fragments were concentrated using Microcon centrifugal columns (Merck) according to the manufacturer's instructions. The benzyldimethylalkylammonium chloride method was used to spread the DNA on the water surface and then loaded on carbon-coated nickel grids. Finally, DNA was coated with 8 nm thick platinum using a high-vacuum sputter EM ACE600 (Leica Microsystems). High-throughput, automated microscopy was performed with a transmission electron microscope (FEI TALOS) setup with a BM-Ceta camera and MAPS 3.17 software. For each experimental condition, at least 70 replication fork intermediates were analysed per experiment, and ImageJ software was used to process and analyse the ssDNA gaps[72].

### Quantification and statistical analysis
Each measurement was taken from an independent biological replicate. Quantifications were performed as noted for each experiment, with no less than three biological replicates for any experiment that was quantified. Statistical analyses were performed as noted, with descriptive analyses, t-test analyses and ordinary one-way ANOVA analyses done through GraphPad Prism software (versions 9 and 10). For descriptive analyses of every quantification mean with standard deviation are shown on graphs. For the ordinary one-way ANOVA analyses, Tukey's multiple comparisons test was used for adjustments, while for the t-test analyses Gaussian distribution was assumed.

### Reporting summary
Further information on research design is available in the Nature Portfolio Reporting Summary linked to this article.

## Data availability
Sequencing data from CRISPR screens generated in this study have been deposited in the NCBI GEO Omnibus database. Raw sequencing data from the CRISPR dropout screen (12 files from Illumina HiSeq 4000 single-end 100 bp sequencing of Brunello sgRNA libraries obtained from day 6 and day 16 of Cas9 induction in wild type and *EXO1* KO cells) and sgRNA counts from MAGeCK analysis have been deposited under accession code GSE255664. Raw sequencing data from CRISPR rescue screens (9 files from Illumina NovaSeq 6000 paired-end 100 bp sequencing of Brunello sgRNA libraries from day 10 of Cas9 induction in *EXO1* KO + sgNT, *EXO1* KO + sg*FANCG* and *EXO1* KO + sg*ZRSR2* cells) and sgRNA counts from MAGeCK analyses have been deposited under accession code GSE255579. Cancer genomics data used in this study were obtained via cBioPortal, with raw data from 32 TCGA studies and the ICGC/TCGA pan-cancer analysis of whole genomes available for download on their website [https://www.cbioportal.org/datasets]. Source data are provided with this paper.

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

## Acknowledgements

The authors would like to thank members of the Boulton lab for helpful discussions and suggestions. The authors would also like to acknowledge members of the science technology platforms teams in The Francis Crick Institute (Advanced Sequencing, Bioinformatics and Biostatistics, Cell Services, Flow Cytometry and High Throughput Screening), as well as the Crick Translation Team for their help in supporting research in the Boulton lab. The authors are grateful to Alan D'Andrea (Dana-Farber Cancer Institute, Harvard Medical School, USA) for providing the PL11 cell line, to Stephen West (The Francis Crick Institute) for FAAP24 antibody, to Weidong Wang (National Institute of Aging, Baltimore, USA) for MHF1 antibody, and to Artios Pharma Ltd for providing FEN1 inhibitor. Work in the S.J.B. lab is supported by The Francis Crick Institute (CC2057), which receives core funding from Cancer Research UK, the Medical Research Council and the Wellcome Trust; European Research Council Advanced Investigator grants (TelMetab, ChrEndProt), a Wellcome Trust Senior Investigator Award, and CRUK RadNet City of London. Work in the A.R.C. lab is supported by NWO (Dutch Research Council) VIDI award (Vidi.193.131) and the Ammodo Science Award.

## Author contributions

M.M. and S.J.B. conceived the study and designed the experiments. S.S.B. helped with the CRISPR screen and performed flow cytometry for cell cycle analysis. R.K. performed and analysed the electron microscopy (EM) experiments. T.T. performed an analysis of metaphase spreads. T.H.S. helped with Cas9 cutting efficiency flow cytometry experiments and with CRISPR rescue screen computational analyses. G.H. prepared reagents for epistasis experiments. R.M. helped with ionising radiation experiments and high-content microscopy. C.F. performed and analysed the fork protection DNA fibre experiments. H.P. performed the computational analysis of sequencing for the CRISPR screen. S.W. performed modelling for the analysis of the apoptosis tracking experiment. M.H. performed imaging and helped with the design and analysis of high-content microscopy experiments. A.R.C. supervised EM and fork protection DNA fibre experiments. P.K. analysed all fork speed DNA fibre experiments. M.M. performed all other experiments with help from V.B., M.P.I., M.L. and N.P. M.M. analysed all data unless noted otherwise.

## Funding

## Competing interests

S.J.B. and M.M. are inventors on a patent WO 2023/047123 ("Method for determining the suitability of an EXO1 inhibitor for the treatment of cancer"). S.J.B. is also co-founder, VP Science Strategy and shareholder at Artios Pharma Ltd. The other authors declare no competing interests.
