## [Peer Review File · Nature Communications]

EXO1 as a therapeutic target for Fanconi Anaemia, ZRSR2 and BRCA1-A complex deficient cancers

Corresponding Author: Dr Simon Boulton

Version 1:

Reviewer comments:

Reviewer #1

(Remarks to the Author)

The authors have now investigated the mechanism of genomic instability observed in EXO1-FANCG double knockouts. They found increased replication fork speed, lower levels of fork reversal, and the presence of single-stranded DNA in electron microscopy micrographs, contributing to genomic instability. Fork protection assays showed that fork degradation in FANCG-depleted cells could be rescued by either EXO1 depletion or MRE11 (Mirin) inhibition. Additionally, the TYMS inhibitor, which rescues the synthetic lethality phenotype of EXO1-FANCG, reduces fork speed, suggesting a way to alleviate replication stress.

I appreciate the authors' efforts to address our comments and perform additional experiments. The manuscript has improved with the new data and analysis. However, I feel the EXO1/ZRSR2 alternative splicing story has only gotten weaker. I would recommend removing it entirely and focusing on the FA pathway interaction, with associated replication stress phenotype.

Below are a major points related to the revision:

- The DNA replication phenotype in EXO1/FANCG cells is intriguing. The difference in fork speed observed by the authors is statistically significant but the effect size is small. As is the rescue with Pemetrexed. The authors should indicate the median fork speed per condition, as used in publication PMID: 29950726, to help readers appreciate the differences. Additionally, for statistical comparison, it should be clarified whether each fiber is used as a data point or if the mean/median of each replicate is compared, to ensure consistency with the observed small differences. A diagram with the staining tracks for Figure 6a would also be helpful. If increased fork speed is the main issue, do the authors believe that any perturbation slowing the fork would rescue the phenotype? This would strengthen their conclusion, as shown with the TYMS inhibitor. Furthermore, what do the authors think about the requirement of MRE11 and EXO1 for fork degradation? Are other nucleases involved? In Figure 6g, depletion of either factor completely rescues fork degradation. How do the authors rationalize this instead of partial rescues? A final model would be useful for discussing the mechanistic understanding of this synthetic relationship.
- The reduced replication fork reversal measurements that are meant to support the increased fork speed are not convincing without statistical testing. As mentioned above, the effect size of EXO1/FANCG dual ko is small but the effect is significant. However, for fork reversal, the effect is small and no significance testing is done. Just looking at the error bars, it seems to me that the effect on fork speed would not be significant. But perhaps the statistical tests would prove me wrong.
- Speaking of statistical testing, the BLISS scores in Supplemental Table 3 nicely show that the EXO1/FANCG interaction is large and statistically significant. However, the EXO1/ZRSR2 interaction is not significant, despite it being the only experiment with 5 replicates. From Supp Table 3, it seems only 6/11 genetic interactions actually exhibit statistically significant synergy rather than additivity. It's good that the authors now focus on EXO1/FANCG, which is indeed a strong and bona fide synergistic interaction.
- The authors no longer focus much on alternative splicing. But I noted that the new data included to support alternative

splicing is not convincing. A bar chart of deltaPSI is not helpful without associated statistical measures from their replicates (e.g. the typical deltaPSI vs p-value volcano plot). The Sashimi plot chosen for FAAP24 shows almost no difference in splicing, which is consistent with the very small deltaPSI for this gene. I'm not sure why the authors chose to show this one, since other genes have greater deltaPSI. Why not show the Sashimi plot for FANCM? As presented, the splicing data is relatively weak and no longer a focus of the story. The authors might consider removing the alternative splicing angle entirely, since it is not a strong aspect of the paper.

Reviewer #2

(Remarks to the Author)

Reviewer #4

(Remarks to the Author)

The authors have satisfactorily addressed the concerns of the reviewers.

Version 2:

Reviewer comments:

Reviewer #1

(Remarks to the Author)

I apologize for the delay in this review. I wanted to discuss my thoughts with the editor before putting pen to paper.

I appreciate the incorporation of some of the suggested changes. They were excellent additions that made the manuscript stronger. However, several of my original concerns were unaddressed and instead argued around. If the alternative splicing story is not removed, it must be made stronger. The editor and I are aligned on this point. As it stands, the splicing section is not convincing.

I won't bother re-stating all the points that the authors chose to argue rather than address. But below are a few of the big ones that still remain. If the authors choose to still leave them unaddressed, they must significantly tone down the wording of the manuscript, which makes their findings seem larger-than-life and incontrovertibly conclusive.

Fork speed: I agreed from the start that the effect is statistically significant. But the difference is very small. The authors need to be more upfront in describing their results.

The authors now indicate median fork speed. But I feel that statistical comparisons of means between biological replicates is appropriate for all samples. For example, this is done in Figures 2g and 4b, c. However, in Figures 6b, g, h and Extended Figure 9, the comparison is done based on individual data points merged between biological replicates. Treating each of the individual fibers (e.g.) across each biological replicate as an individual sample leads to hundreds to thousands of comparisons but without false discovery rate correction. The reported means for fork speed in the same samples differ between Figures 6b and 6h, suggesting experimental variance. This variance must be properly analyzed and interpreted. Given that the conclusions drawn from these experiments are central to the proposed mechanistic model, it is crucial that the authors discuss this thoroughly. If possible, alternative interpretations or scenarios should also be included in the discussion.

The splicing effects are quite minor. I appreciate the authors' statement that the "the alternative splicing defect is stochastic". But citation 46 (prior work showing that ZRSR2) shows quite a few strong and specific effects on U12-type splicing introns only. Are the authors now suggesting that ZRSR2 specifically affects some transcripts (prior work) but stochastically affects splicing for their genes of interest? A volcano plot of per-transcript Δ PSI vs statistical significance should be included in the main text, as this is standard in the field. This should be shown next to one or more Sashimi plots (e.g. currently in Extended Figure 7). Let the readers draw their own conclusions about the extent of the effect for the targets called out as important. A Sashimi plot for FANCM should be one of the genes shown. FANCM appears to be one of the most significantly affected genes, and given its potential contribution to the observed phenotype, it is essential to report this. A physical validation assay—such as RT-PCR targeting the mis-spliced region—should be included to confirm the RNA-seq findings and provide stronger support for the proposed splicing alterations.

Finally, the authors have not performed any rescue experiments for ZRSR2. This omission should be acknowledged in the manuscript, as without such experiments, the possibility of off-target effects remains.

Reviewer #2

(Remarks to the Author)

Version 3:

Reviewer comments:

Reviewer #1

(Remarks to the Author)

The new edits have addressed all of my concerns. The experiments are well performed and the paper is tightly and clearly written. I congratulate the authors on their excellent work. And I would look forward to a follow-up story about the potential splicing effects if further investigation and additional results in this direction shows this would be warranted.

Reviewer #2

(Remarks to the Author)

We would like to thank the reviewers for their constructive suggestions for improving the manuscript. As advised, we have focused on one specific synthetic lethal interaction (between EXO1 and FANCG) as an exemplar and have investigated the underlying mechanistic basis. Prompted by our original observation that *EXO1-FANCG* DKO cells exhibit elevated replication stress markers, we now show the following:

- DNA fibre analysis reveals that *EXO1-FANCG* DKO cells exhibit increased replication fork speeds.
- DNA fibre analysis of *EXO1-FANCG* DKO cells treated with high dose of PARP inhibitor olaparib (which has been shown previously to increase fork speed) further increases replication fork speed in DKO cells, suggesting a mechanism for additive sensitivity we have shown in the manuscript previously.
- Direct visualisation of replication intermediates by electron microscopy revealed a defect in replication fork reversal in *EXO1-FANCG* DKO cells and an accumulation of post-replicative single stranded gaps, which is absent in single KO cells.
- *FANCG* KO cells exhibit increased fork degradation, which is suppressed by loss of EXO1 or mirin treatment.
- We show that synthetic lethality of *EXO1-FANCG* DKO cells can be rescued by TYMS inhibitor treatment (*TYMS* was identified as one of the top hits in our CRISPR rescue screen), which restored the fork speeds back to wild type levels.

These new data reveal an unexpected role for EXO1 and FANCG in promoting replication fork reversal, which when compromised leads to accelerated replication fork speeds and post-replicative ssDNA gaps, which drive genome instability and loss of viability.

In addition, we have also significantly expanded the section on ZRSR2, including new alternative splicing analysis, epistasis analysis between ZRSR2 and the FA pathway and rescue experiments with TYMS inhibition. Details of the revisions can be found below.

Reviewers' Comments:

Reviewer #1 and Reviewer #2:

Remarks to the Author:

In this manuscript, Maric et al. found that the viability of some cancer cells is dependent on Exo1 in DepMap and employed a CRISPR-Cas9-based genome-wide screen in eHAP cells to identify synthetic lethality. This approach led to the identification of numerous candidates, which the authors binned into potential functional pathways. Among these candidates, they focus their analysis on cells lacking an intact FA pathway or ZRSR2. Both avenues validated, and the authors attributed these phenotypes to genomic instability issues leading to apoptosis. Interestingly, they identified ZRSR2 as crucial for the alternative splicing of numerous genes, including some within the FA/BRCA pathway that leads to FA-like sensitivities. To identify possible rescue alleles, the authors conducted a synthetic viability screen in double knockout ZRSR2/Exo1 and FANCG/Exo1 cells. They discovered several genes involved in DNA replication stress, as well as TYMS in both screens.

I found the Exo1 synthetic lethal concept to be interesting. The paper is a bit split in its goals. On the one hand, it makes several predictions for potential therapies and attempts to be a roadmap for using Exo1 as a target in FA/BRCA/ZRSR2 cancers. On the other hand, it proposes mechanisms by which the synthetic lethality exerts their function. While both are interesting avenues, I feel that neither is sufficiently fleshed out enough to be of great interest. Mechanistic readers will be left wondering how things really work, and therapeutic readers will not be convinced that these synthetic hits are truly good starting points for preclinical studies. I recommend that the authors choose one major area, leave the other where it stands in the paper, and do much more to flesh out the chosen portion. A few suggestions are below, but the authors could take other routes. The important thing is that they really do something significant in one area, rather than halfway in two areas.

On the therapeutic side:

While Exo1 is synthetic lethal with FA/ZRSR2 in a few routinely used cancer cell lines, the clinical relevance is not clear. The authors could expand their investigations to many more cell backgrounds. Best would be PDX and xenograft models to determine the effect of appropriate intervention in cells from patients with the relevant genotype. e.g. the effect Exo1 knockout in BRCA1 patient cells during xenotransplant in mice. Or to model chemical intervention, cisplatin treatment in Exo1 knockout cells after xenotransplantation. The latter would be interesting, especially since some of the dose response curves in Figures 1, 3, and 4 do not show a very strong differential from wildtype.

Thank you for the constructive suggestions. Reviewers 1, 2, 4 and the editor advised us to develop either the therapeutic or mechanistic aspects of the work for the revision. We opted to further expand the mechanistic aspect of our work by investigating the underlying causes of one of the synthetic lethal interactions. In revised Figure 6 we have investigated replication dynamics in *EXO1-FANCG* double KO (DKO) cells, which revealed an unexpected increase in replication fork speed in comparison to either wild type or single KO cells. Such a phenotype has been described before in cells treated with PARP inhibitors, which has been attributed to a defect in replication fork reversal. In collaboration with Arnab Ray Chaudhuri's laboratory, electron microscopy analysis of replication intermediates revealed a defect in replication fork reversal in *EXO1-FANCG* DKO cells as well as accumulation of single stranded DNA gaps behind the fork. We further demonstrate that the resulting high replication fork speed is the driver of synthetic lethality in the DKO cells as TYMS inhibitor Pemetrexed, which we identified as a top hit in the rescue CRISPR screen, rescues the loss of viability in *EXO1-FANCG* and *EXO1-ZRSR2* DKO cells (now in Figure 5 of revised manuscript) by reducing replication fork speed to wild type levels in *EXO1-FANCG* DKO cells. These new data reveal an unexpected role for EXO1 and FANCG in promoting replication fork reversal, which when compromised leads to accelerated replication fork speeds and post-replicative ssDNA gaps, which drive genome instability and loss of viability. The rescue of this phenotype by TYMS inhibitor satisfactorily links the synthetic lethal and rescue screen data in the paper. In addition, we have explored the genetics of *ZRSR2* KO synthetic lethality with EXO1 loss and epistasis with the Fanconi anaemia pathway, revealing that loss of FAAP24 does not further increase loss of viability upon EXO1 loss induction. Furthermore, our new data show that loss of FAAP24 does not further exacerbate the cisplatin sensitivity of

ZRSR2 KO. These results are now a part of Extended Data Figure 7 connected to revised Figure 3. We hope the reviewers will concur that these new data provide an important addition to our study.

On the mechanism side:

It is unclear to me if FA/*ZRSR2* is truly synthetic with Exo1. Many of the colony survival quantifications show that single perturbations are somewhat toxic on their own. It is hard to see if the double perturbations are beyond additive. This must be clarified to call this a synthetic interaction.

We have now provided further clarification on the genetic interactions and can show that they are synergistic rather than additive. We have now included a table with theoretical BLISS scores alongside actual viability scores for every biological replicate of clonogenic assays for each of the validated synthetic lethal interactions in the manuscript (now Supplementary Table 3). These calculations demonstrate that the observed genetic interactions are indeed synergistic when comparing actual to theoretical values. Therefore, comparisons of these individual paired values still allow for the 'synthetic lethality' term to be used for these interactions and for those to be deemed as synergistic.

Since Exo1 loss shows deficiencies in HR, MMC repair, and IR repair (all hallmarks of the FA pathway), how do the authors reconcile their results that FA pathway loss even further contributes to cell death? This is relevant for my question above about additivity rather than synergy.

As mentioned in the first paragraph of our response to Reviewers 1 and 2, we now have evidence of synergy between EXO1 and Fanconi Anaemia pathway with regards to fork reversal deficiency, as shown in Figure 6. Furthermore, additional data now included in Figure 6 describes a defect in fork degradation upon HU treatment, which is dependent on MRE11 and EXO1 processing. Our data shows that while fork degradation is present in FANCG-deficient cells, forks in *EXO1-FANCG* DKO cells cannot undergo degradation, which likely explains the synergy rather than additivity.

There is no mechanistic data to show how *ZRSR2* contributes to splicing of several FA genes. Exactly which genes are affected is unclear, as is the magnitude of the effect. This is missing from both the main figures and the supplement. The RNA-seq data is never truly shown beyond GO-term analysis and a very disappointing venn diagram in Supplemental Figure 7. This is not OK for modern transcriptomic or splicing analysis. I would greatly appreciate plots shown transcript abundance, deltaPSI for exons, sashimi plots of differentially spliced genes, and so on. Once convinced that splicing is truly strongly affected, a next step would be to figure out how *ZRSR2* manages to be selective for the FA pathway, since it affects several of these genes.

The original manuscript was submitted with a supplementary table containing data from the analysis (including gene identities and relevant values). However, we acknowledge that this data was not visible enough and consequently we now present the mRNA-Seq data differently in the revised manuscript. As part of the revised Figure 3 we provide plots of delta Psi values for each of the relevant category of genes that were found to be alternatively spliced in *ZRSR2* inducible KO cells 4 days after KO induction (Fanconi Anaemia genes, other known genome stability factors and factors

which were identified to be synthetic lethal with EXO1 loss in the genome-wide CRISPR dropout screen and which are also alternatively spliced in *ZRSR2* inducible KO cells). In Figure 3 we also further highlight the position of all alternatively spliced genes within the *EXO1* KO CRISPR dropout screen to accurately represent the overlap between the two datasets. In associated Extended Data Figure 7 we also show a representative Sashimi plot of one of the alternatively spliced FA genes in *ZRSR2* KO (FAAP24), for which we later show epistasis with *ZRSR2* KO with regards to EXO1 synthetic lethality and cisplatin sensitivity (Extended Data Figure 7). However, we would like to point out that our data and our conclusions do not suggest that *ZRSR2* is selective for FA pathway genes as many other genes are also affected.

The synthetic viability screen is an interesting idea, and the authors say that replication stress is the reason for rescue in double Exo1/FA knock out cells. But many of the genes found are common essential (e.g. GINS2, CDC6). It possible that their synthetic interaction occurs when multiple genes that make a cell sick just cannot make it even more sick (this is a common false positive in finding synthetic viable hits).

The Common TYMS hit is interesting. But mechanism proving their replication stress hypothesis is almost entirely missing. Examination of replication fork speed, protection, and so on need to be shown in their double knockout cells.

We thank the reviewers for this suggestion. As outlined above, we have now explored these phenotypes in depth using DNA fibre experiments and direct visualisation of replication intermediates by electron microscopy. In short, our new data shown in revised Figure 6 reveals an unexpected increase in replication fork speed in *EXO1-FANCG* DKO cells, which is further increased with a PARP inhibitor (which likely explains the additional sensitivity described in the original submission, now in Figure 5 of the revised manuscript), deficiency in fork reversal observed via EM, increase in single stranded DNA gaps observed via EM and rescue of fast replication fork speed phenotype by TYMS inhibitor Pemetrexed, leading to viability rescue/tolerance.

As a minor point, I would suggest the authors show their microscopy images in single-channel format rather than merged with DAPI channel; then it would be easier to assess the focus numbers from the representative images.

We thank the reviewers for this suggestion. However, we have chosen to represent the images as merged format to accurately reflect the exact images and channel exposures captured by the high-content system which were used for the automated analysis with Harmony software, for which we show the quantification here.

Reviewer #3:

Remarks to the Author:

In this manuscript, the authors performed CRISPR screen to identify genes that show synthetic lethality with loss of EXO1. As expected, the authors uncovered many genes involved in various DNA repair pathways. They further validated several genes in FA pathway and BRCA1-A complex. Additionally, they showed that loss of *ZRSR2*, a gene involved in splicing, affected splicing events including those genes in FA pathway. Moreover, they showed that loss of genes they identified not only sensitized cells to EXO1 depletion/inhibition and also sensitized cells to its combination with olaprib or radiation.

The data present in general support their conclusion. However, the scientific advance is very limited. EXO1 is known to participate in multiple DNA repair pathways. Thus, it is not surprising that many DNA repair genes displayed synthetic lethality with EXO1 loss. In addition, there are many reports suggesting that defects in RNA splicing and/or splicing factors would lead to increased DNA damage and make these cells sensitive to DNA damaging agents, probably due to R loop formation and/or other mechanisms.

With all due respect to the reviewer, we disagree with the comment that our findings are of 'limited scientific advance'. As well as anticipated genetic interactions reported previously that validated our approach, we report multiple novel synthetic lethal interactions of EXO1 among DDR factors, such as with FA genes and BRCA1-A, which we followed up due to the potential utility of this finding as a novel targetable vulnerability in cancer. Importantly, we also report on synthetic lethal interactions with EXO1 loss for genes that have not been previously connected to genome stability maintenance, such as ZRSR2, which is a known and *bona fide* cancer driver gene. We dispute the statement of Reviewer 3 that our findings on the genome instability phenotype of ZRSR2 loss as being one of 'many' such splicing factors. ZRSR2 is amongst the overall top synthetic lethal hit in our genome-wide dropout screen in *EXO1* KO cells and is the only spliceosome factor that scored as significantly synthetic lethal with EXO1 loss. We now provide an additional panel in Figure 3 to emphasize this point, depicting positions of other spliceosome factors in our genome-wide CRISPR dropout screen in *EXO1* KO cells, none of which score as synthetic lethal. If this were a general phenomenon associated with splicing then we would have expected to detect multiple splicing factors as synthetic lethal with EXO1 loss in our screen, which is not the case. With regards to the statement from Reviewer 3 that "many reports suggesting that defects in RNA splicing and/or splicing factors would lead to increased DNA damage and make these cells sensitive to DNA damaging agents, probably due to R loop formation and/or other mechanisms", we have now added an additional volcano plot highlighting position of R-loop processing factors in the *EXO1* KO CRISPR dropout screen. If aberrant splicing and R-loop formation were a particular vulnerability in *EXO1* KO cells, then we would expect to observe numerous R-loop processing factors as synthetic lethal with EXO1. This is not the case, as shown in Extended Data Figure 7.

Our goal was to set out to identify genetic vulnerabilities in *EXO1* KO cells, which could be potentially exploited in cancer treatment. The importance of our study is exemplified by but not limited to our finding that ZRSR2, a known cancer-driver gene homozygously deleted in ~6% of all cancer and is SL with EXO1 loss (<https://doi.org/10.1038/s41586-020-1969-6>; <https://doi.org/10.1038/s41588-019-0562-0>). Our finding that ZRSR2 clusters with FA genes in the sensitivity arm of our EXO1 SL screen, impacts the alternative splicing of subset of FA genes and phenocopies FA deficient cells with reduced FANCD2-Ub, increased radial chromosome formation and sensitivity to ICL-inducing agents (hallmarks of FA) provides an explanation for why ZRSR2 KO (like FA deficient cells) is SL with EXO1 loss. In the revised paper, we also show that the synthetic lethality of *EXO1-FANCG* and *EXO1-ZRSR2* DKO cells are both rescued by TYMS inhibition, which further strengthens our claim of phenotypic similarity of ZRSR2 and FANC mutations in this context.

Finally, in response to the comments of other reviewers, we have provided additional mechanistic insight into the synthetic lethality of *EXO1-FANCG*. In the revised paper our new data show 1) elevated replication fork speeds, defective fork reversal and accumulation of ssDNA gaps behind forks in *EXO1-FANCG* DKO cells, and 2) suppression of replication fork speeds and rescue of viability by TYMS inhibitor (identified in our rescue screens). These data provide an unexpected and significant mechanistic advance that explains the synthetic lethality observed between EXO1 and FANCG.

Additional concerns:

1) Deletions and/or mutations in many DNA repair genes listed are quite low, which have not yet been experimentally validated.

These numbers are derived from published studies available through cBioPortal.

2) Increased sensitivity to combination treatments has been reported by many investigators. However, the major concern is whether any combination truly increase anti-tumor efficacy in cancer patients without increasing toxicity. Unfortunately, this key question cannot be addressed in vitro especially when there is no EXO1 specific inhibitor.

We agree with the reviewer that any combination therapeutic approach highlighted in this study would need to be thoroughly tested according to standards for any new drug and its combinations. However, we believe that this does not negate the value of testing possible combination opportunities in a genetic system, as well as informing on the types of exogenous DNA lesions that cannot be resolved in double KO cell lines as data relevant for mechanistic understanding of synthetic lethality. Certainly, our study will provide a framework to test the utility of an EXO1 inhibitor alone and in combination, if and when such a molecule becomes available.

3) As stated in the manuscript, many genetic drivers of EXO1 synthetic lethal interactions were essential genes. Moreover, TYMS has been reported previously by several groups to be synthetic lethal with loss of DDR genes (for example, please see DOI: [10.1186/s12943-021-01405-8](https://doi.org/10.1186/s12943-021-01405-8)).

We agree with the Reviewer's statement, but we do not understand how this is relevant. BRCA1 and BRCA2 are essential genes, but their loss in cancer is tolerated by loss of p53. Evidently TYMS loss or inhibition is tolerated and improves the survival of *EXO1-FANCG* and *EXO1-ZRSR2* DKO cells, but not in single KO cells, which is the new data that we now show in the revised manuscript (revised Figure 5). We also now provide a molecular explanation for this effect with new data in Figure 6. We show by DNA fibre assay that *EXO1-FANCG* DKO cells display increased replication fork speed, which can be rescued with the addition of TYMS inhibitor Pemetrexed, which has been shown in the literature to decrease replication fork speed in a dose-dependent manner (DOI: [10.1016/j.molcel.2024.04.004](https://doi.org/10.1016/j.molcel.2024.04.004)).

Reviewer #4:

Remarks to the Author:

In this manuscript, Maric and colleagues conduct genome-wide CRISPR-KO screens in EXO1 KO eHAP and HeLa cells to identify EXO1 synthetic lethal interactions. The authors show that EXO1 deficiency results in synthetic lethality in combination with loss of genes of the Fanconi anemia (FA) pathway and the BRCA1-A complex. Furthermore, the authors identify novel genetic interactions between EXO1 and genes not previously implicated in genome maintenance, such as ZRSR2 and CDK11B. Overall, the genes that exhibit synthetic lethal interactions with EXO1 are homozygously deleted in ~8% of all cancers. The authors also show that the synthetic lethal interactions between EXO1 and some of the above genes are dependent on the catalytic activity of EXO1, suggesting that EXO1 catalytic inhibitors could be developed for cancer treatment.

Overall, the findings of this study are well-presented and of interest to the DNA damage and repair community. Determining the mechanistic bases of the described synthetic lethal interactions and defining their potential clinical relevance would be important to strengthen the manuscript, as discussed below.

Major points

1) The authors should validate the identified synthetic lethal interactions in relevant cancer models mutated in the genes that exhibit synthetic lethal interactions with EXO1. These studies could also include analysis of the DepMap dataset to determine whether mutations and/or changes in the expression of the above genes correlate with EXO1 dependency.

We agree with the Reviewer's interest in further investigating these synthetic lethal interactions in a context of relevant cancer models. However, in the interest of the cohesiveness of this manuscript and the stronger interest from all reviewers to explore a synthetic lethal interaction mechanistically, we have focused on the mechanistic aspects of an exemplar synthetic lethal interaction (*EXO1-FANCG*) in the revised manuscript.

2) The authors should provide further mechanistic insights into the identified synthetic lethal interactions. Which of the multiple DNA repair pathways in which EXO1 operates are critical for the observed EXO1 interactions? Do genes that operate in those pathways also show synthetic lethal interactions with FA genes, BRCA1-A complex genes and/or ZRSR2? Which functions of FA genes, BRCA1-A complex genes and ZRSR2 are required to suppress the synthetic lethality observed in EXO1 KO cells?

This is an interesting suggestion but clearly beyond the scope of this paper, which aimed to identify targetable vulnerabilities with EXO1. This could easily take 18-24 months to complete.

3) It would be important to determine whether ZRSR2 displays an epistatic relationship to FA genes. In particular, what is the effect of ZRSR2 loss in FA-deficient cells in response to EXO1 loss or treatment with crosslinking agents?

We thank the Reviewer for this suggestion. We have now performed these experiments to confirm epistasis between ZRSR2 and FA pathway with regards to both EXO1 synthetic lethality and cisplatin sensitivity. The new data, which is now a

part of Extended Data Figure 7 (associated with main Figure 3), confirms epistasis of ZRSR2 and FA via a clonogenic assay through the loss of FAAP24, which is one of the genes we validated as synthetic lethal with EXO1 loss and for which multiple discreet splicing defects have been determined in the mRNA-Seq data. Furthermore, we also show in Extended Data Figure 7 that cisplatin sensitivity of ZRSR2 KO is epistatic with FAAP24.

4) Further validation and characterization of the hits from the suppressor screens would be important to provide insights into the underlying causes of the synthetic lethal interactions of EXO1 with FANCG and ZRSR2.

Please see response to the last major point of Reviewers 1 and 2 comments above.

Minor points

1) Fig.1b. In most panels, EXO1 deficiency is associated with a mild but consistent impairment of cell proliferation (see Fig.2b-c; Fig.4b-c; Ext. Fig.1b-d; Ext. Fig.2c-e; Ext. Fig.3a-i; Ext. Fig. 4a-i; Ext. Fig.5a-b; Ext. Fig.6a-h, k-n). Is the non-significant effect on cell proliferation induced by EXO1 loss in Fig.1b due to the control sgRNA? It would be helpful if the authors could comment on these findings or repeat the experiment in Fig. 1b.

The mild effect on cell proliferation is likely due to EXO1 loss on its own. Data presented in Figure 1b was collected for KO cell lines which do not have any integrated sgRNAs. A similar effect is also observed for HeLa Kyoto KO cell lines, where we see a comparable proliferation phenotype for KO clones that were individually generated with different sgRNAs, which would eliminate any possible off-target effect of a guide. Furthermore, the majority of cancer cell lines investigated in Project Achilles (DepMap) show a mild cell proliferation impairment with EXO1 loss (attaching a screenshot from DepMap for the gene effect data for EXO1).

Dependent Cell Lines ⓘ

CRISPR (DepMap Public 24Q2+Score, Chronos):
55/1150

STRONGLY SELECTIVE ⓘ

RNAi (Achilles+DRIVE+Marcotte, DEMETER2):
0/710

2) Fig. 1i-j. Are the presented screen data obtained using multiple EXO1 KO clones? If not, it would be helpful to repeat the screens with an additional EXO1 KO clone to confirm the hits identified in the screen.

The CRISPR dropout screen was performed with a single eHAP iCas9 EXO1 KO clone (clone 11) with a biological triplicate at the level of transduction with the Brunello lentiviral library. Clone 11 was thoroughly validated functionally prior to the screen

together with other eHAP iCas9 *EXO1* KO clones, showing very similar effects of different DNA damaging agents and inhibitors of DDR factors on the viability of clones (Figure 1c-e, Extended Data Figure 1f-k). Furthermore, we highlighted and validated genes that were shown to be synthetic lethal with *EXO1* loss even in an evolutionary distant model organism such as budding yeast (Extended Data Figure 2a-g), further demonstrating the reproducibility of data obtained from the screen across organisms. Validation of our findings was also conducted a HeLa Kyoto iCas9 *EXO1* KO clone generated with a different sgRNA, which targeted a different exon from the sgRNA used for clone 11 generation in eHAP iCas9 cell line. As shown in Extended Data Figure 4a-l, validation in HeLa Kyoto iCas9 *EXO1* KO clone 2.19, as well as reciprocal validation in FANCC-deficient cell line PL11 (Extended Data Figure 4j-n) confirmed the reproducibility of the main findings derived from the screen. Finally, we complemented eHAP iCas9 *EXO1* KO clone with WT and catalytic dead *EXO1* and could show that the WT rescued *EXO1* SL interactions with FANCG and ZRSR2 KOs whereas the catalytic dead *EXO1* did not. This demonstrates that these SL interactions are specific to *EXO1* and require *EXO1* catalytic activity. Given the extent of the approaches used to validate our main findings, including validation in a second cell line, repeating a screen in another eHAP clone is an unnecessary expense and is not justifiable for this study.

3) Fig. 1i-j and 4i-k. It would be useful to show the screen data also on a scatter dot plot with WT cells on the X axis and *EXO1* KO cells on the Y axis (Fig. 1) or *EXO1* KO cells on the X axis and *EXO1*/FANCG or *EXO1*/ZRSR2 double KO cells on the Y axis (Fig 4). In Fig. 4, it would be helpful to also show whether sgRNAs targeting FANCG and ZRSR2 display expected phenotypes in the screen. If so, to which extent sgRNAs targeting genes in the FA pathway and genes regulated by ZRSR2-dependent splicing events score differently from FANCG and ZRSR2 sgRNAs in *EXO1* KO vs double KO cells?

We thank the reviewer for the suggestion to highlight FA genes in the rescue screens and we attach the volcano plots for their attention. In their respective rescue screens both of sets of Brunello library sgRNAs for either FANCG or ZRSR2 appear on the 'rescue' side of the screen, which we interpret is due to the significantly lower MOI than the non-Brunello sgRNA for each gene (MOI >1) that was integrated on top of the Brunello library sgRNAs for the rescue screen (meaning they are more neutral in double KO than in single *EXO1* KO). Importantly, we observe other FA genes on the 'rescue' side of both screens, strongly suggestive of epistasis. Interestingly, we observe both FANCM and FAAP24 on the 'rescue' side of the *EXO1*-ZRSR2 DKO screen, which were both among alternatively spliced factors in ZRSR2 KO mRNA-Seq.

EXO1 KO sg*FANCG* vs sgNT

EXO1 KO sgZRSR2 vs sgNT

We also appreciate the suggestion for a different form of plotting screen data. However, this kind of plotting is not compatible with MAGECK analysis as it would require normalisation for sgRNAs, which is less robust and has lower statistical power than MAGECK analysis, and which is likely to be skewed towards the biological repeat that is chosen to be plotted rather than taking the variability between biological repeats into account during the analysis for the significance of the output.

4) Extended Data Fig. 3. The same representative image for "EXO1 KO + sgNT" appears to have been used twice in panels c and d.

We thank the reviewer for this observation, which we have now corrected and replaced the representative image in Extended Data Figure 3d.

5) Extended Data Fig. 4j-m. It would be helpful to show whether FANCC reconstitution suppresses the growth defect caused by EXO1 loss in PL11 cells.

We thank the Reviewer for this suggestion. We have now added data with FANCC reconstitution, which reconfirmed our previous result. As part of revised Extended Data Figure 4 associated with main Figure 2, we show that re-expressing wild type FANCC rescues the ability of PL11 cells to monoubiquitylate FANCD2. In the same experiment we induce loss of EXO1 via lentiviral transductions of Cas9 and sgRNA against EXO1 and still observe the significant delay in the exponential growth phase of FANCC-

deficient PL11 cells. Importantly, this is not the case for FANCC-reconstituted PL11 cell line.

6) Extended Data Fig. 7. The authors should conduct a more comprehensive analysis of gene expression and splicing alterations in ZRSR2 KO cells. Are there specific patterns of splicing events affected by ZRSR2 loss?

As suggested by this and Reviewers 1 and 2, we now show a more thorough analysis of the mRNA-Seq dataset, as elaborated in the response to Reviewers 1 and 2.

Reviewer's Comments:

Reviewer #1 (Remarks to the Author)

The authors have now investigated the mechanism of genomic instability observed in EXO1-FANCG double knockouts. They found increased replication fork speed, lower levels of fork reversal, and the presence of single-stranded DNA in electron microscopy micrographs, contributing to genomic instability. Fork protection assays showed that fork degradation in FANCG-depleted cells could be rescued by either EXO1 depletion or MRE11 (Mirin) inhibition. Additionally, the TYMS inhibitor, which rescues the synthetic lethality phenotype of EXO1-FANCG, reduces fork speed, suggesting a way to alleviate replication stress.

I appreciate the authors' efforts to address our comments and perform additional experiments. The manuscript has improved with the new data and analysis. However, I feel the EXO1/ZRSR2 alternative splicing story has only gotten weaker. I would recommend removing it entirely and focusing on the FA pathway interaction, with associated replication stress phenotype.

We thank the reviewer for acknowledging our efforts in addressing theirs and the other reviewer's comment. With respect to the EXO1/ZRSR2 data in our study, we respectfully disagree with the reviewer and concur with the editor who has asked us not to remove these data from the study.

Below are a major points related to the revision:

- The DNA replication phenotype in EXO1/FANCG cells is intriguing. The difference in fork speed observed by the authors is statistically significant but the effect size is small. As is the rescue with Pemetrexed.

We would like to highlight that the increase in replication fork speed in EXO1-FANCG DKO has been seen consistently in multiple independent experiments, including the original experiment as well as in control conditions for experiments with Pemetrexed and Olaparib conducted for the revision all of which we repeated at least 3 times. Furthermore, the original DNA fibre experiments were performed by PK, while MM performed the Pemetrexed and Olaparib experiments, as noted in the author contributions section. Therefore, we would consider these results highly reproducible and very robust across experiments and performed by different people.

The authors should indicate the median fork speed per condition, as used in publication PMID: 29950726, to help readers appreciate the differences. Additionally, for statistical comparison, it should be clarified whether each fiber is used as a data point or if the mean/median of each replicate is compared, to ensure consistency with the observed small differences.

Thanks for highlighting this point. In the main figures we have used each DNA fibre as a data point, with medians with interquartile ranges now highlighted on the graphs relating to fork speed experiments (in red) and numbers for fork speed means noted below the X-axis.

A diagram with the staining tracks for Figure 6a would also be helpful.

We have now added a staining diagram in Figure 6a next to the representative images with a representative track (from 'WT + sgNT' image, as noted in the legend) that was used for measurement of total fork speed, together with a diagram for how the measurement was done.

If increased fork speed is the main issue, do the authors believe that any perturbation slowing the fork would rescue the phenotype? This would strengthen their conclusion, as shown with the TYMS inhibitor.

This is entirely consistent with our data. We refer the reviewer to our rescue screens, which identified many other replication fork factors as significant hits. We have highlighted some of the replication factors that came out as hits in both screens in the text, including GINS2 (constitutive part of replicative helicase CMG, which is the central part of the replisome) and CDC6 (essential DNA origin firing factor and obligatory part of the pre-replication complex). However, further analysis of these factors is not possible as they are essential. Instead, we chose to follow up TYMS as it was one of the top hits in both rescue screens, is non-essential and because it can be inhibited by Pemetrexed. Our data with Pemetrexed treatment validated the genetic rescue screen result with TYMS depletion and showed that this rescue correlates with a reduction in replication fork speeds in the *EXO1-FANCG* DKO cells. Conversely, olaparib treatment of *EXO1-FANCG* DKO cells further increases fork speed and leads to additional cell killing, which further substantiated our hypothesis that increased replication fork speeds are ultimately responsible for the *EXO1-FANCG* synthetic lethality. The other reviewers concur with our conclusion.

Furthermore, what do the authors think about the requirement of MRE11 and EXO1 for fork degradation? Are other nucleases involved? In Figure 6g, depletion of either factor completely rescues fork degradation. How do the authors rationalize this instead of partial rescues?

We cannot exclude the possibility of other nucleases acting up or downstream of EXO1 in the context of FA core complex deficiency; however, further analysis would require a genetic screen of all other nucleases, which is clearly outside the scope of this paper. It should be noted that other nucleases have been shown to be involved in fork degradation in different contexts (a review from Morris lab summarises involvement of different nucleases depending on the context of different deficiencies; PMID: 32653304). In one such example MRE11, EXO1 and MUS81 have been shown to be involved in fork degradation in the context of BRCA2 deficiency (PMID: 29038425, *Nature Communications*). As dissection of the fork degradation mechanism in the context of BRCA deficiency was the focus of their entire paper, this further emphasises the depth of mechanistic insight our study has offered in the revised version. Furthermore, the complete rescue of fork degradation with either MRE11 or EXO1 is entirely consistent with published data on short and long range resection of DSBs, with the former dependent on MRE11 and the latter on EXO1.

A final model would be useful for discussing the mechanistic understanding of this synthetic relationship.

We have included a model as requested, which is now in Figure 7 in the manuscript.

- The reduced replication fork reversal measurements that are meant to support the increased fork speed are not convincing without statistical testing. As mentioned above, the effect size of *EXO1/FANCG* dual ko is small but the effect is significant. However, for fork reversal, the effect is small and no significance testing is done. Just looking at the error bars, it seems to me that the effect on fork speed would not be significant. But perhaps the statistical tests would prove me wrong.

We have now included the statistics for the electron microscopy experiments as per previous studies, such as in PMID: 34555355. We have performed a two-way ANOVA test for both analyses to compare 4 samples across untreated and HU-treated conditions.

For quantification of reversed forks we have used Dunnett's multiple comparisons. Among untreated samples we do not observe any significance. However, HU-treated samples are significant between WT and each sample (WT+sgNT vs. WT+sgFANCG: 0.0030; WT+sgNT vs. *EXO1* KO+sgNT: 0.0304; and WT+sgNT vs. *EXO1* KO+sgFANCG: 0.0006). This also correlates with the rest of the observations, where sgFANCG cells show lower fork reversal due to fork degradation, while *EXO1* KO + sgFANCG DKO cells do not reverse the forks, hence the increase in fork speed.

For the quantification of ssDNA molecules we also used Dunnett's multiple comparisons to compare the entire group (0, 1, 2, >2) across the samples. Specifically, we compared the change in trend between WT and each sample, by taking all the ssDNA molecules as one group. Although there is a significance between multiple pairs in 0 and 1 ssDNA molecules, only the WT+sgNT vs. *EXO1* KO+sgFANCG comparison shows significance in all the molecules in both untreated and HU treated samples (we have plotted statistical comparison for 2 ssDNA molecules). This suggests that there is an increase in the exposed ssDNA molecules at the stalled forks, as the forks are not being reversed.

- Speaking of statistical testing, the BLISS scores in Supplemental Table 3 nicely show that the *EXO1*/*FANCG* interaction is large and statistically significant. However, the *EXO1*/*ZRSR2* interaction is not significant, despite it being the only experiment with 5 replicates. From Supp Table 3, it seems only 6/11 genetic interactions actually exhibit statistically significant synergy rather than additivity. It's good that the authors now focus on *EXO1*/*FANCG*, which is indeed a strong and bona fide synergistic interaction.

While we appreciate the acknowledgement of the strength of *EXO1-FANCG* DKO synthetic lethal interaction, we do not discredit the rest of the interactions we have presented in the manuscript. We are copying below the excerpt from the table relating to actual viability scores and BLISS scores for *EXO1-ZRSR2* synthetic lethal interaction to showcase this.

double knockout combination with EXO1 loss	biological repeat number	actual viability score for repeat (normalised against WT +sgNT)	BLISS score for repeat (based on actual scores from respective single KOs)	P value of unpaired t-test
sg ZRSR2	1	0.119601	0.32404719	0.0546
	2	0.22409	0.40524445	
	3	0.449153	0.97081777	
	4	0.259386	0.59001267	
	5	0.202797	0.40894443	

Analysis of each pair of actual vs theoretical value, shows that theoretical value is always much higher for all repeats. We acknowledge that the distribution of these values is also wide (with actual scores ranging from 0.11 to 0.44, and BLISS scores ranging from 0.32 to 0.97), and as a result of that the P value is just below the threshold of significance in an unpaired t-test. While we fully agree with the reviewer on focusing on the strongest interactions, we do not want to dismiss some of the very relevant biological interactions due to a wider data distribution.

- The authors no longer focus much on alternative splicing. But I noted that the new data included to support alternative splicing is not convincing. A bar chart of deltaPSI is not helpful without associated statistical measures from their replicates (e.g. the typical deltaPSI vs p-value volcano plot). The Sashimi plot chosen for FAAP24 shows almost no difference in splicing, which is consistent with the very small deltaPSI for this gene. I'm not sure why the authors chose to show this one, since other genes have greater deltaPSI. Why not show the Sashimi plot for FANCM? As presented, the splicing data is relatively weak and no longer a focus of the story. The authors might consider removing the alternative splicing angle entirely, since it is not a strong aspect of the paper.

For the alternative splicing analysis we have used Whippet algorithm that calculates cumulative deltaPSI values for an alternatively spliced gene. For at least two out of three biological repeats such a gene has at least 10% of transcripts alternatively spliced. It is important to acknowledge that the alternative splicing defect is stochastic with respect to a specific gene and only occurs at a certain frequency. Results are displayed through a table with a single deltaPSI value, single probability value, single complexity calculations and single value for entropy. With regards to the choice of the gene for the Sashimi plot, we chose to focus on FAAP24 as this was an FA factor that we followed up and showed was epistatic with *ZRSR2* KO cells with respect to sensitivity to cisplatin and synthetic lethality with *EXO1* loss. In contrast to FAAP24 or the other two FANCM complex cofactors *MHF1/APITD1* and *MHF2/STRA13*, FANCM has additional roles in genome stability maintenance besides its function in the FA pathway, which would have complicated epistasis experiments. Despite the stochastic nature of the alternative splicing defect observed in *ZRSR2* KO cells, these discreet changes cumulatively lead to a robust defect in FA pathway activation in *ZRSR2* KO. The latter is unambiguous with *FANCG* KO and *ZRSR2* KO exhibiting the following phenotypic similarities: synthetic lethality with *EXO1* loss, sensitivity to PARPi or cisplatin, accumulation of DNA damage markers and radial chromosomes. As such, we respectfully disagree with the reviewer and as advised by the editor we will retain the *ZRSR2* data in the paper.

Point by point response to reviewer

Reviewer #1 (Remarks to the Author):

I apologize for the delay in this review. I wanted to discuss my thoughts with the editor before putting pen to paper.

I appreciate the incorporation of some of the suggested changes. They were excellent additions that made the manuscript stronger. However, several of my original concerns were unaddressed and instead argued around. If the alternative splicing story is not removed, it must be made stronger. The editor and I are aligned on this point. As it stands, the splicing section is not convincing.

After further consultation with the editor regarding the splicing dataset, we have now removed the entire splicing data from the manuscript. As advised by the editor, we have amended Figure 3 to add all the data from Figure 4, which contained the *ZRSR2* KO phenotypic data. These data have not been questioned by any of the reviewers and they demonstrate that *FANCG* KO and *ZRSR2* KO exhibit the following phenotypic similarities: synthetic lethality with *EXO1* loss, spontaneous accumulation of DNA damage markers, radial chromosomes and defective *FANCD2* monoubiquitylation in response to damage with DNA crosslinkers, as well as sensitivity to PARPi or cisplatin. We further show that *ZRSR2* and *FAAP24* are epistatic with respect to cisplatin sensitivity, as well as synthetic lethality with *EXO1* loss. In new Figure 4 (previous Figure 5), we further show that the synthetic lethality of *EXO1-FANCG* DKO and *EXO1-ZRSR2* DKO cells depends on the catalytic activity of *EXO1* and are both suppressed by *TYMS* inhibition, demonstrating additional phenotypic similarities between *FANCG* KO and *ZRSR2* KO cells. We have included a summary of this in the revised discussion.

I won't bother re-stating all the points that the authors chose to argue rather than address. But below are a few of the big ones that still remain. If the authors choose to still leave them unaddressed, they must significantly tone down the wording of the manuscript, which makes their findings seem larger-than-life and incontrovertibly conclusive.

Fork speed: I agreed from the start that the effect is statistically significant. But the difference is very small. The authors need to be more upfront in describing their results.

The authors now indicate median fork speed. But I feel that statistical comparisons of means between biological replicates is appropriate for all samples. For example, this is done in Figures 2g and 4b, c. However, in Figures 6b, g, h and Extended Figure 9, the comparison is done based on individual data points merged between biological replicates. Treating each of the individual fibers (e.g.) across each biological replicate as an individual sample leads to hundreds to thousands of comparisons but without false discovery rate correction.

We thank the reviewer for their suggestions. We agree that expressing means is a more appropriate statistical measure for comparisons and we are now showing the statistical analysis considering only the means of biological replicates in datasets

relating to fork speed in what are now panels of Figure 5b and h and Extended Data Figure 9. These analyses demonstrate that increased fork speed in *EXO1-FANCG* DKO cells is statistically significant in each of these experimental sets of data when comparing means of biological replicates, which we have also noted on each of the fork speed plots. Importantly, the new analysis does not change our conclusion regarding the rescue of replication fork speed phenotype with Pemetrexed (Figure 5h). However, the trend we see with the further increase of replication fork speed with olaparib treatment is not statistically significant with the new analysis (shown in amended Extended Data Figure 9). This is likely to be due to the cells having reached near to the maximum replication speed achievable in this cellular system. As such, we have changed how we refer to this in the text. Regarding the text describing the statistically significant increase of fork speed in unchallenged conditions, we acknowledge that increases of fork speed seen in these three independent datasets are modest and we have now made changes in the discussion to reflect this. These modest but significant increased fork speeds together with the fork reversal defect, accumulation of ssDNA at forks as shown by both EM and through DDR markers, lack of fork resection (for which fork reversal is a prerequisite) all point to dysregulation of replication dynamics in *EXO1-FANCG* DKO cells.

The reported means for fork speed in the same samples differ between Figures 6b and 6h, suggesting experimental variance. This variance must be properly analyzed and interpreted.

We have included the following section in the Methods paragraph on DNA fiber analysis: “Subtle variance between fork speed experiments could be due to assays being performed by two different researchers over an 18 month period with different numbers of samples (experiments in Figure 5b in comparison with experiments in Figure 5h and Extended Data Figure 9). Variance could also result from differences in batches of reagents used, differences in media composition and modest changes in atmospheric oxygen conditions. Despite variance in absolute numbers between several experiments, statistical significance was observed confirming the robustness of the findings.”

Given that the conclusions drawn from these experiments are central to the proposed mechanistic model, it is crucial that the authors discuss this thoroughly. If possible, alternative interpretations or scenarios should also be included in the discussion.

We have now updated our model both in the text and in amended Figure 6 by discussing the possibility of homologous recombination and/or single-strand annealing deficiency contributing to defective repair of DSBs derived from ssDNA gaps.

The splicing effects are quite minor. I appreciate the authors' statement that the “the alternative splicing defect is stochastic”. But citation 46 (prior work showing that ZRSR2) shows quite a few strong and specific effects on U12-type splicing introns only. Are the authors now suggesting that ZRSR2 specifically affects some transcripts (prior work) but stochastically affects splicing for their genes of interest? A volcano plot of per-transcript Δ PSI vs statistical significance should be included in the main text, as this is standard in the field. This should be shown next to one or more Sashimi plots (e.g. currently in Extended Figure 7). Let the readers draw their own conclusions about the extent of the effect for the targets called out as important. A Sashimi plot for

FANCM should be one of the genes shown. FANCM appears to be one of the most significantly affected genes, and given its potential contribution to the observed phenotype, it is essential to report this. A physical validation assay—such as RT-PCR targeting the mis-spliced region—should be included to confirm the RNA-seq findings and provide stronger support for the proposed splicing alterations.

These data have been removed at the request of the reviewer.

Finally, the authors have not performed any rescue experiments for ZRSR2. This omission should be acknowledged in the manuscript, as without such experiments, the possibility of off-target effects remains.

To minimise potential off target effects, it is generally acceptable to either conduct analyse in several independently derived KO clones (and in our case, in 2 different cell lines) or complementation of one KO clone. We opted for the former as it was important to demonstrate the effects we observed were not restricted to a single cell line. While we agree that complementation experiments would have further strengthened our conclusions, the possibility of off-target effects in this case is minimal as 1) we show that loss of ZRSR2 is synthetic lethal with EXO1 loss in a genome-wide CRISPR screen where ZRSR2 was a top hit observed with 4 different sgRNAs (Fig.1j, Supplementary Table 2), 2) we validated that ZRSR2 is synthetic lethal with EXO1 loss in two different cell lines: eHAP iCas9 (Fig.3a-c, Extended Data Fig.6i) and HeLa Kyoto iCas9 (Extended Data Fig.6k-l), and 3) we tested 3 independently derved KO clones of ZRSR2 in eHAP iCas9 cells for sensitivity with cisplatin and olaparib (Fig.3k-l), with each KO showing similar levels of sensitivity in dose response curves (Extended Data Fig.7b).

Reviewer #2 (Remarks to the Author):
